# Mild replication stress causes premature centriole disengagement via a sub-critical Plk1 activity under the control of ATR-Chk1

Devashish Dwivedi [1,2], Daniela Harry[1,2] & Patrick Meraldi [1,2] ✉

A tight synchrony between the DNA and centrosome cycle is essential for genomic integrity. Centriole disengagement, which licenses centrosomes for duplication, occurs normally during mitotic exit. We recently demonstrated that mild DNA replication stress typically seen in cancer cells causes premature centriole disengagement in untransformed mitotic human cells, leading to transient multipolar spindles that favour chromosome missegregation. How mild replication stress accelerates the centrosome cycle at the molecular level remained, however, unclear. Using ultrastructure expansion microscopy, we show that mild replication stress induces premature centriole disengagement already in G2 via the ATR-Chk1 axis of the DNA damage repair pathway. This results in a sub-critical Plk1 kinase activity that primes the pericentriolar matrix for Separase-dependent disassembly but is insufficient for rapid mitotic entry, causing premature centriole disengagement in G2. We postulate that the differential requirement of Plk1 activity for the DNA and centrosome cycles explains how mild replication stress disrupts the synchrony between both processes and contributes to genomic instability.

The centrosomes in animal cells are the major microtubule organising centres (MTOC) that regulate the interphase microtubule network and control the poles of the mitotic spindle during cell division. Centrosomes also integrate and coordinate multiple signalling pathways involved in the regulation of cell polarity, migration, development, and fate[1,2]. Centrosomal dysfunctions may result in developmental disorders or cancer[2–4]. Each centrosome consists of two tightly associated and orthogonally oriented barrel-shaped centrioles that are 500 nm long and 250 nm wide, called mother and daughter centrioles. These two centrioles are surrounded by a protein-rich pericentriolar matrix (PCM). In dividing animal cells, the two centrioles within centrosomes duplicate once per cell cycle in a process that is tightly controlled in space and time[5]. The centriole duplication cycle starts with centriole disengagement during the telophase of the previous cell cycle. Indeed, a steric blockade inhibits the formation of new procentrioles as long as both centrioles within the centrosome are in tight orthogonal association with each other[6,7]. Centriole disengagement is under the control of the mitotic kinase Plk1 and the protease Separase[8,9], with Plk1 targeting by phosphorylating the pericentriolar protein pericentrin for Separase-dependent cleavage[8–11]. As centrioles disengage, they lose their orthogonal orientation and steric hindrance, licensing them for duplication during the next S phase[12]. The duplicated centrosomes, each containing a pair of engaged centrioles, separate at the mitotic onset to form the poles of the bipolar spindle before being segregated into the two daughter cells during anaphase.

Under normal conditions, the centriole duplication cycle proceeds synchronously with the cell cycle as both DNA and centrosomes are licensed for replication at the end of mitosis, replicated in S phase and segregated to the two daughter cells during mitosis[5]. Consistently, both processes share multiple common regulatory proteins[13]. Any dysregulation in the centriole duplication cycle can lead to the formation of abnormal spindles in mitosis, in particular multipolar spindles, which are commonly observed in many cancers[14]. Even though cancer cells possess centrosome clustering mechanisms to convert

[1]Department of Cell Physiology and Metabolism, Faculty of Medicine, University of Geneva, 1211 Geneva 4, Geneva, Switzerland. [2]Translational Research Centre in Onco-hematology, Faculty of Medicine, University of Geneva, 1211 Geneva 4, Geneva, Switzerland. ✉e-mail: patrick.meraldi@unige.ch

multipolar spindles into bipolar spindles, transient abnormal spindles will favour erroneous chromosome attachments to spindle microtubules, increasing the probability of chromosome instability (CIN) and aneuploidy[15].

We recently reported that mild replication stress induced by nanomolar doses of the DNA-polymerase inhibitor Aphidicolin causes premature centriole disengagement in mitosis, disrupting the synchrony between the centrosome and the cell cycle. Premature centriole disengagement in cells under replication stress resulted in transient multipolar spindles that often led to chromosome segregation errors and chromosome instability in anaphase[16]. Replication stress is recognised as any cellular condition in which DNA replication is slowed down or hampered, a condition which is already present in many pre-cancerous lesions[17]. It can result in the formation of double or single DNA strand breaks (DSBs/SSBs), which activate the damage repair (DDR) kinases Ataxia telangiectasia mutated (ATM; in case of DSBs) or Ataxia Telangiectasia and Rad3-related protein (ATR; in case of SSBs) to delay mitotic onset[18,19]. The link between the DDR pathway and the centriole duplication cycle, however, remains so far unclear. Here, we combined small molecule-based inhibition against different cell cycle and DDR regulators and protein depletions with ultrastructure expansion microscopy (U-ExM) to unravel the molecular signalling pathway involved in premature centriole disengagement under mild replication stress conditions. Our results show that centriole disengagement depends on the ATR-Chk1 axis of the DDR pathway and that it is caused by creating a sub-critical level of Plk1 activity that suffice to drive premature centriole disengagement via the Separase-pericentrin cleavage mechanism in G2, but are insufficient to promote efficient mitotic entry itself.

## Results

### Mild replication stress induces premature centriole disengagement in G2

To investigate how mild replication stress disrupts the synchrony between the cell- and centriole duplication cycle, we worked with untransformed human retinal pigment epithelial (hTERT-RPE1) cells immortalised with human telomerase. These cells have functional cell-cycle checkpoints, low basal incidences of chromosome segregation errors and a normal centriole duplication cycle. To induce mild replication stress, we applied low doses (400 nM) of the DNA polymerase Aphidicolin for 16 h, a condition known to induce premature centriole disengagement in mitosis[16]. To study the origin of premature centriole disengagement, we used ultrastructure expansion microscopy (U-ExM) to test whether low doses of Aphidicolin already induced centriole disengagement in the S- or G2-phase. This methodology allowed us to overcome the resolution limit of conventional light microscopy and to determine whether the centriole pair within a centrosome was engaged or not (Fig. 1A). Centrioles that were closely associated (R: the ratio between the distance between the mother and daughter centriole over the length of mother centriole R ≤ 1) and in perfect orthogonal orientation (angle between the centriole pair 90 ± 5°) were considered engaged; centriole pairs that did not meet these criteria were classified as disengaged (Fig. 1A).

We first analysed the centriole configuration in unsynchronised cells. In S-phase cells, which contained short procentrioles (<50% length of the parental centriole), all centriole pairs were engaged whether cells were treated with DMSO (negative control) or 400 nM Aphidicolin (Fig. 1B, C). In contrast, $45.30 \pm 4.81\%$ of the G2 cells (procentriole length >50% of parental centriole) treated with 400 nM Aphidicolin displayed disengaged centrioles vs $2.63 \pm 1.85\%$ of the DMSO-treated G2 cells (Fig. 1D–F; Supplementary Fig. S1A). Our previous study indicated that Cdk1 activity was required during the low Aphidicolin treatment to induce premature centriole disengagement in mitosis[16]. Consistently, Cdk1 inhibition using the small molecule inhibitor RO3306[20] also suppressed centriole disengagement in

Aphidicolin-treated G2 cells ($18.62 \pm 6.54\%$) while having no effect on its own (Fig. 1D–F). To confirm this cell cycle dependence, we synchronised cells at the G1/S transition with the CDK4/6 inhibitor Palbociclib[21] before releasing them in media containing DMSO or 400 nM Aphidicolin for 4 h (S phase) and 8 h (G2 phase). Our quantification indicated that low doses of Aphidicolin only led to disengaged centrioles in cells released for 8 h ($33.13 \pm 2.27\%$), confirming that mild replication stress only induces premature centriole disengagement in G2 cells (Fig. 1G, H; Supplementary Fig. S1B).

The two centriole pairs in a G2 cell are not identical. Due to the semi-conservative mechanism of centrosome duplication, one centrosome is always older[5]. To evaluate if this age asymmetry affects the probability for centrioles to disengage, we co-stained centrioles with Centrobin, a centriolar protein localising in G2 to both daughter centrioles and the parental centriole of the young centrosome[22]. Our quantification indicated that the proportion of disengaged centrioles in old or young centrosomes was not statistically different ($10.99 \pm 2.59\%$ vs. $13.76 \pm 0.22\%$; Fig. 1I, J). Moreover, we found a high proportion of cells in which both centriole pairs were disengaged ($15.64 \pm 4.41\%$), implying that the disengagement of both centriole pairs is controlled by a common upstream factor. We conclude that mild replication stress already induces centriole disengagement in G2, independent of the centriole age.

### ATR-Chk1 regulates premature centriole disengagement under mild replication stress conditions

Replication stress induces activation of the DDR pathway[23] and delays mitotic entry by engaging the G2/M checkpoint[24]. We therefore mapped out which part of the DNA damage machinery might regulate centriole disengagement in G2 under mild replication stress conditions. We first inhibited the upstream DNA damage repair kinases ATR (Ataxia telangiectasia and Rad3-related) and ATM (ataxia-telangiectasia mutated) in Aphidicolin-treated cells using the selective small molecular inhibitors ETP46464 (ATR inhibitor) and KU55933 (ATM inhibitor)[25,26]. Inhibiting ATR activity suppressed the Aphidicolin-induced centriole disengagement in the G2 phase ($7.40 \pm 3.70\%$ vs. $45.41 \pm 3.03\%$ with Aphidicolin alone; Fig. 2A, B). Inhibiting ATM activity under similar replication stress conditions did not prevent premature centriole disengagement in G2 ($40.97 \pm 3.92\%$; Fig. 2A, B). Finally, treatment with ATM ($9.92 \pm 1.10\%$) or ATR ($11.08 \pm 8.19\%$) inhibitors alone led to a very mild increase in centriole disengagement (DMSO: $3.13 \pm 0.08\%$). We conclude that premature centriole disengagement in G2 under mild replication stress conditions depends on ATR but not on ATM kinase activity.

To initiate DNA repair and prevent mitotic entry in cells having damaged DNA, ATR activates the protein kinase Chkl, while ATM activates the Chk2 kinase. We therefore next suppressed the activity of Chk1 with two independent inhibitors, LY2603618 (Chk1i-1) and PF-477736 (Chk1i-2)[27,28] and Chk2 using small molecule inhibitors (Chk2i)[29] in Aphidicolin-treated cells and monitored the centriole engagement status. None of the inhibitors affected centriole engagement when administered alone ($3.54 \pm 0.27\%$: Chk1i-1, $3.92 \pm 0.37\%$: Chk1i-2, $3.73 \pm 0.38\%$: Chk2; Supplementary Fig. S1C, D). Chk1 inhibition, however, suppressed premature centriole disengagement in G2 in Aphidicolin-treated cells ($20.04 \pm 3.79\%$: Chk1i-1 and $19.10 \pm 4.01\%$: Chk1i-2), whereas Chk2 inhibition had no effect ($43.28 \pm 2.95\%$ vs $42.23 \pm 3.46\%$ in Aphidicolin treatment alone; Fig. 2C, D). We conclude that the premature centriole disengagement in G2 induced by mild replication stress depends on the ATR-Chk1 arm of the DNA damage repair pathway.

### Premature centriole disengagement under replication stress depends on Wee1 and Cdk1/Cyclin-A

The ATR-Chk1 arm of the DNA damage repair pathway activates the Wee1 kinase, which phosphorylates Cdk1 at Tyrosine-15 to prevent

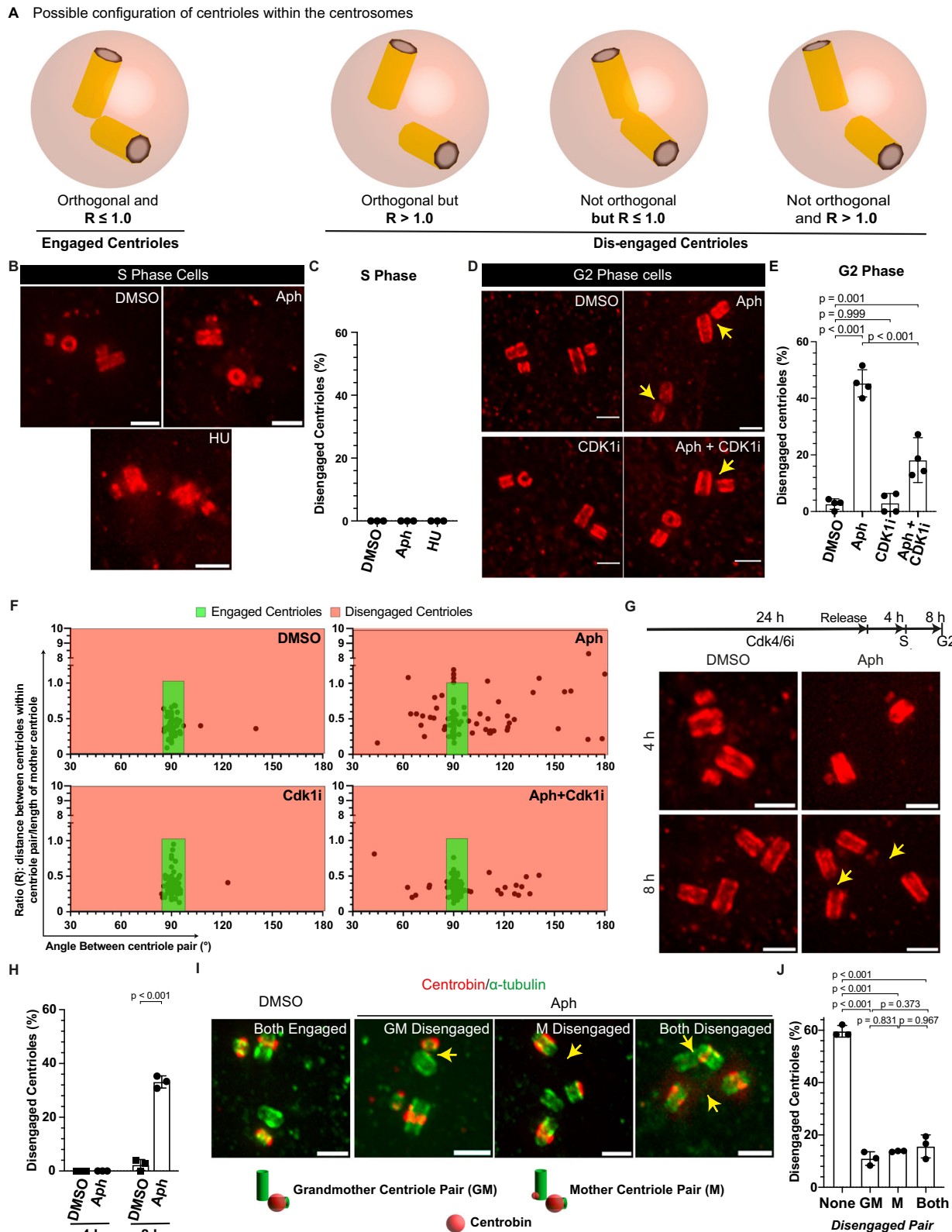

**A** Possible configuration of centrioles within the centrosomes

Orthogonal and **R ≤ 1.0** | Orthogonal but **R > 1.0** | Not orthogonal but **R ≤ 1.0** | Not orthogonal and **R > 1.0**

**Engaged Centrioles** | **Dis-engaged Centrioles**

mitotic entry and prolongs G2[30,31]. To evaluate to which extent Wee1 activity and/or a prolonged G2 phase is required for a premature centriole disengagement, we inhibited Wee1 using the small molecule inhibitors PD0166285 (Wee1i-1) and MK1775 (Wee1i-2)[32,33]. Since such a treatment dramatically shortens G2, we could not analyse the centriole configuration in G2 cells by U-ExM. Instead, we used live-cell imaging to detect transient multipolar mitotic spindles that arise after

Aphidicolin treatment[16]. While RPE1 cells expressing EB3-GFP (mitotic spindle marker) and H2B-mCherry (DNA marker) underwent normal cell divisions after DMSO treatment (0.52 ± 0.57% multipolar spindles; Supplementary Movie 1), Aphidicolin treatment significantly favoured the formation of transient multipolar spindles during early mitosis (15.45 ± 1.19%; Fig. 3A, B, Supplementary Fig. S2A; Supplementary Movie 2), as previously observed[16]. Inhibition of Wee1 on its own did

**Fig. 1 | Mild replication stress causes premature centriole disengagement in the G2 phase. A** Possible centriole configurations in G2 centrosomes to identify centrosomes with disengaged centrioles. **B** U-ExM images of centrioles in S phase RPE1 cells stained against α-tubulin and treated with indicated drugs. **C** Quantification of the percentage of S-phase cells with disengaged centrioles in their centrosomes ($N = 3$ independent experiments, $n = 86$, 79 and 81 cells for DMSO, Aph and HU, respectively). **D** U-ExM images of centrioles in G2 phase RPE1 cells stained against and α-tubulin treated with indicated drugs/inhibitors. **E** Quantification of the percentage of G2 phase cells with disengaged centrioles in their centrosomes. Each dot in the plot represents one centriole pair ($N = 4$ independent experiments, $n = 111$, 128, 101 and 111 cells in DMSO, Aph, Cdk1i and Aph+Cdk1i, respectively: $p$-values from two-tailed Sídak test). **F** Dot plot showing the distribution of centriole orientation with respect to the ratio of centriole distance to mother centriole length. Each dot in the plot represents one centriole pair ($N = 4$ independent experiments, $n = 111$, 128, 101 and 111 cells in DMSO, Aph, Cdk1i and

Aph+Cdk1i, respectively). **G** U-ExM images of centrioles in RPE1 cells at the indicated time after release from G1 arrest and stained against α-tubulin and treated with indicated drugs. **H** Quantification of the percentage of cells with disengaged centrioles at the indicated time after Cdk4/6i release ($N = 3$ independent experiments, $n = 4$ h DMSO: 95, 8 h DMSO: 86, 4 h Aph: 103 and 8 h Aph: 87 cells: $p$-values from two-tailed Sídak test.) **I** U-ExM images of centrioles in G2 phase RPE1 cells stained against α-tubulin (green) and Centrobin (red) and treated with indicated drugs. **J** Quantification of the percentage of cells with disengaged centrioles in their old and new centrosomes ($N = 3$ independent experiments, $n = 85$ and 109 cells in DMSO and Aph, respectively; $p$-values from two-tailed Sídak test). The cells in Aph were further categorised based on the age of the disengaged centriole pair, as indicated. The yellow arrowheads in respective U-ExM images indicate a disengaged centriole pair. Data presented as mean ± SD. Scale bars = 0.5 μm. Source data for all graphs are provided as a Source Data file.

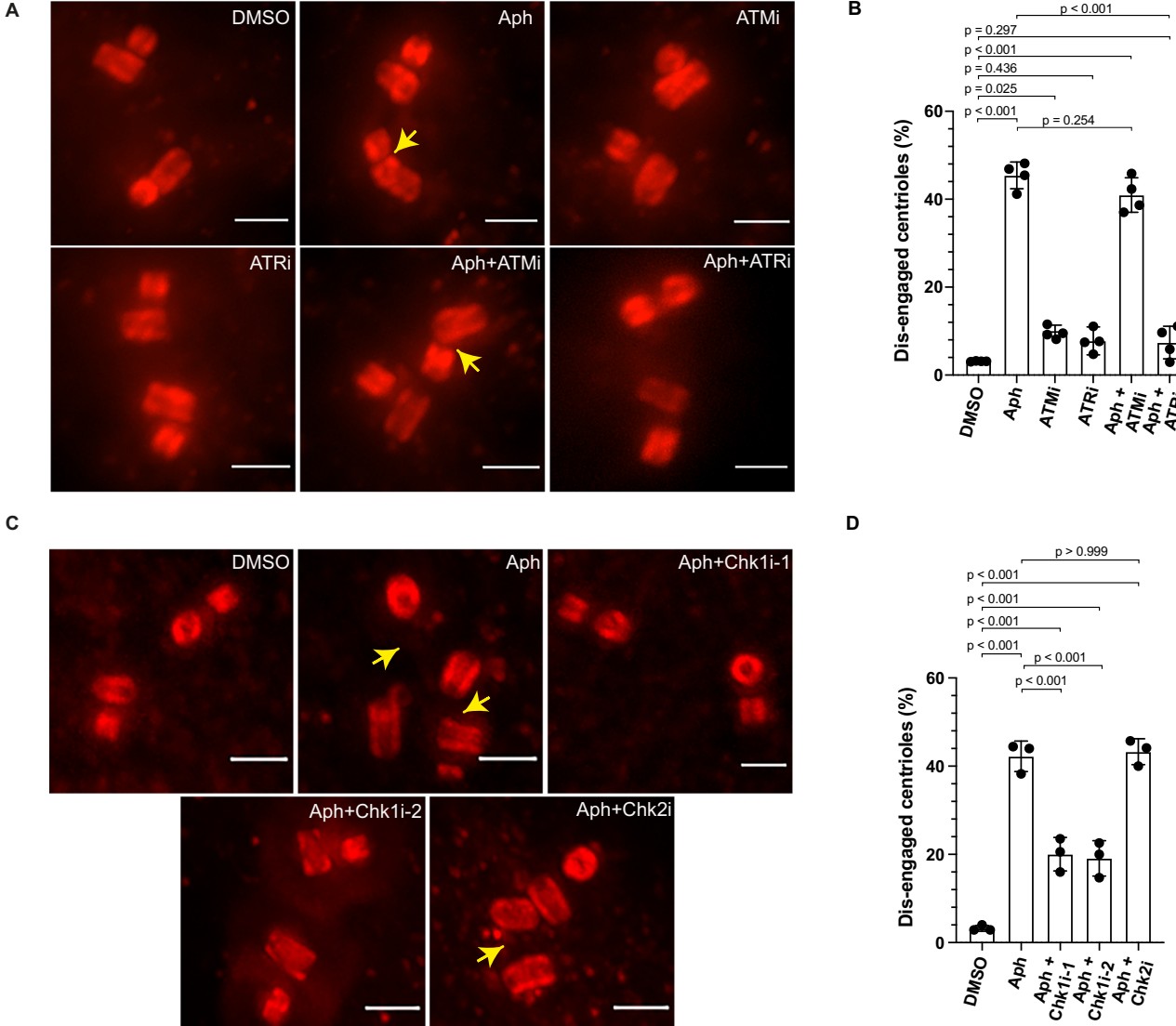

**Fig. 2 | Premature centriole disengagement during mild replication stress depends on the ATR/Chk1 axis of the DDR Pathway. A** U-ExM images of centrioles in G2 Phase RPE1 cells, treated with indicated drugs/inhibitors targeting the DNA damage repair pathway. **B** Quantification of the percentage of G2 phase cells with engaged centrioles in their centrosomes ($N = 3$ independent experiments, $n = 96$, 93, 70, 71, 76 and 92 cells in DMSO, Aph, ATMi, ATRi, Aph+ATMi and Aph+ATRi, respectively; $p$-values from two-tailed Sídak test). **C** U-ExM images of

centrioles in G2 Phase RPE1 cells, treated with indicated drugs/inhibitors targeting Chk1 or Chk2. **D** Quantification of the percentage of G2 phase cells with engaged centrioles in their centrosomes ($N = 3$ independent experiments, $n = 95$, 95, 92, 90 and 94 cells in DMSO, Aph, Aph+Chk1i-1, Aph+Chk1i-2 and Aph+Chk2i, respectively). Data presented as mean values ± SD. ($p$-values from two-tailed Sídak test). Scale bars = 0.5 μm. Source data for all graphs are provided as a Source Data file.

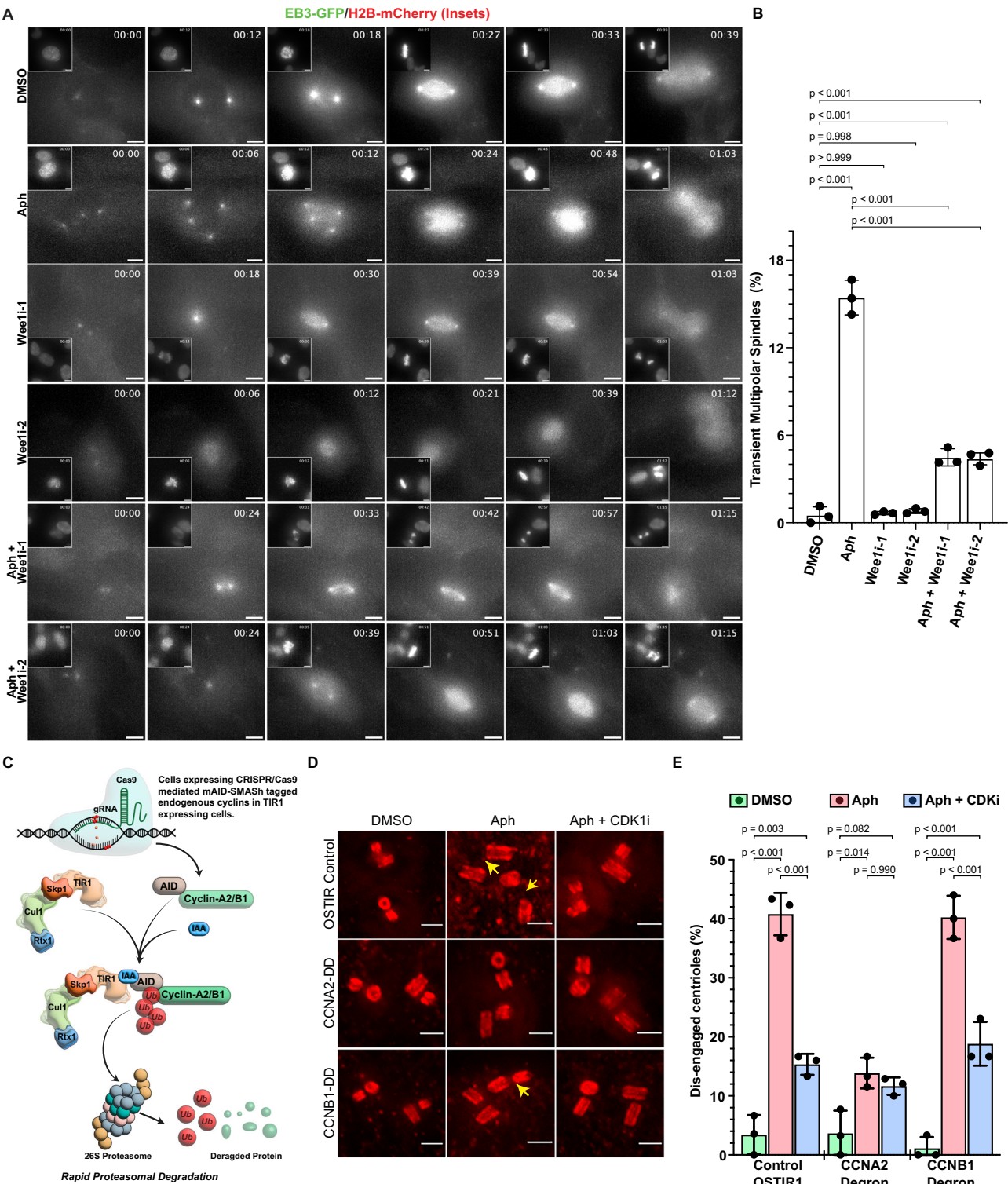

**Fig. 3 | Premature centriole disengagement depends on Wee1 kinase and Cdk1/Cyclin-A2. A** Live-cell time-lapse images of RPE1: EB3-eGFP/H2B-mCherry cells treated with indicated drugs/inhibitors. EB3-GFP channel is shown as a large image and the corresponding H2B-mCherry (DNA) channel is shown as insets. Scale bars = 5 μm. **B** Quantification for the proportion of mitotic cells displaying transient multipolar spindles upon mitotic entry in (A) (*N* = 3 independent experiments, *n* = 485, 137, 462, 385, 333 and 276 cells in DMSO, Aph, Wee1i-1, Wee1i-2, Aph +Wee1i-1 and Aph+wee1i-2, respectively: *p*-values from two-tailed Sídak test). **C** Schematic representation of auxin-induced cyclin degradation of Cyclin-A2 and B1 in RPE1 cells. **D** Expansion microscopy images of centrioles in G2 Phase RPE1

OSTIR1, Cyclin-A2 double degron (CCNA2-DD) and Cyclin-B1 double degron (CCNB2-DD) cells after inducing degron expression and treated with indicated drugs/inhibitors. **E** Quantification of percentage of G2 phase cells with disengaged centrioles in their centrosomes [*N* = 3 independent experiments, *n* = 88 (OSTIR:DMSO), 86 (OSTIR:Aph), 85 (OSTIR:Aph+Cdk1i), 88 (CCNA2-DD:DMSO), 86 (CCNA2-DD:Aph), 86 (CCNA2-DD:Aph+Cdk1i), 85 (CCNB1-DD:DMSO), 75 (CCNB1-DD:Aph) and 70 (CCNB1-DD:Aph+Cdk1i) cells]. Data presented as mean values ± SD. Data presented as mean values ± SD. (*p*-values from two-way ANOVA and Dunn's multiple comparison test). Scale bars = 0.5 μm. Source data for all graphs are provided as a Source Data file.

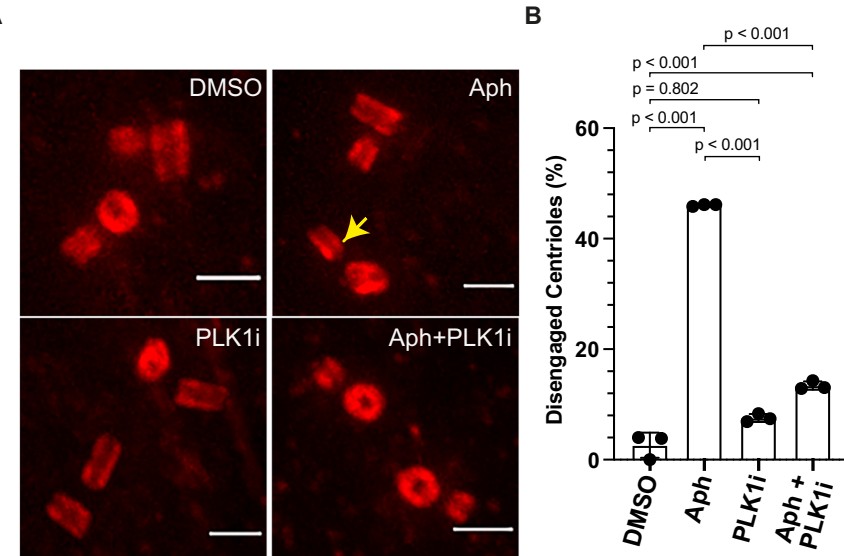

**Fig. 4 | Premature centriole disengagement requires Plk1 activity. A** Expansion microscopy images of centrioles in G2 phase RPE1 cells treated with indicated drugs/inhibitors. **B** Quantification of the percentage of G2 phase cells with engaged centrioles in their centrosomes (*N* = 3 independent experiments, *n* = 79, 76, 80 and 82 cells in DMSO, Aph, Plk1i, and Aph+Plk1i, respectively). Data presented as mean values ± SD. (*p*-values from two-tailed Sídak test). Scale bars = 0.5 μm. Source data for all graphs are provided as a Source Data file.

not change the frequency of transient multipolar spindle (0.66 ± 0.11%: Wee1i-1 and 0.80 ± 0.16%: Wee1i-2; Supplementary Movies 3 and 4) but suppressed this phenotype in Aphidicolin-treated cells (4.49 ± 0.59%: Wee1i-1 and 4.38 ± 0.41%: Wee1i-2; Fig. 3A, B, Supplementary Fig. S2A, Supplementary Movies 5 and 6). We conclude that Wee1 activity is required for premature centriole disengagement after mild replication stress.

Even though Cdk1 inhibition is the main action of Wee1[31], premature centriole disengagement requires Cdk1 activity[16]. This activity is driven in the late G2 phase by two cyclins: Cyclin-A2 and Cyclin-B1, which are expressed at similar levels at this stage. To determine which cyclin drives centriole disengagement in G2, we used established RPE1 cells expressing auxin-inducible degron (AID) endogenously tagged Cyclin-A2 or Cyclin-B1[34], which allow rapid degradation of the tagged protein by the ubiquitin/proteasome system (Fig. 3C and validation of depletion in Supplementary Fig. S2B, C). Degradation of Cyclin-A2 or Cyclin-B1 alone led to normal rates of premature centriole disengagement in G2 when compared to the parental cell line expressing only the E3-ubiquitin ligase OSTIR (3.41 ± 3.34%: OSTIR; 3.64 ± 3.86%: Cyclin-A2-AID and 1.11 ± 1.92%: Cyclin-B1-AID; Fig. 3D, E). However, premature centriole disengagement induced by Aphidicolin in G2 was suppressed upon Cyclin-A2 degradation, but not by Cyclin-B1 degradation (13.85 ± 2.60%: Cyclin-A2, Cyclin-B1: 40.22 ± 3.67%, 40.77 ± 3.59%: OSTIR; Fig. 3D, E). In line with a Cdk1/CyclinA-2 dependent centriole disengagement, Cdk1 inhibition only suppressed premature centriole disengagement in control (15.33 ± 1.76%) and Cyclin-B1-AID (18.80 ± 3.70) cells, but had no further effect in Cyclin-A2-AID cells (11.63 ± 1.49%; Fig. 3D, E). We conclude that premature centriole disengagement in G2 is driven by Cdk1/CyclinA-2.

### Replication stress results in intermediate Plk1 activity

One of the prominent downstream effectors of Cdk1/Cyclin-A is the protein kinase Plk1[35,36]. Plk1 is required for the physiological centriole disengagement[11,37,38] in telophase and the premature centriole disengagement in mitosis after replication stress[16]. Moreover, Plk1 overexpression can induce centriole disengagement[37]. We therefore inhibited Plk1 in Aphidicolin-treated cells using a small molecule inhibitor, BI-2536[39] and quantified centriole disengagement in G2. We found a strong suppression of centriole disengagement (13.41 ± 0.76%) when compared to cells treated with Aphidicolin alone (46.05 ± 0.19%) and no significant change in the absence of Aphidicolin (3.77 ± 0.36%: Plk1 inhibition vs 2.62 ± 2.27%: DMSO; Fig. 4A, B). We conclude that premature centriole disengagement in G2 also depends on Plk1 activity.

Overall, our results pointed to a paradox: premature centriole disengagement in G2 depends on the ATR-Chk1-Wee1 pathway that limits Cdk1 and Plk1 activity, yet at the same time, it depends on Cdk1/CyclinA2 and Plk1. We therefore hypothesised that mild replication stress might lead to a partial Plk1 activity that would be sufficient to drive centriole disengagement but not sufficient for rapid mitotic entry. To test this hypothesis, we quantified Plk1 activity using a Fös-ter's Resonance Energy Transfer (FRET) Plk1 activity sensor based on a c-Jun peptide (Plk1 substrate) coupled to CFP and YFP[40]. Phosphorylation of this peptide induces a conformational change that decreases the FRET efficiency allowing to quantify Plk1 activity. Quantitative immunofluorescence of RPE1 cells expressing the Plk1-FRET sensor indicated high FRET (YFP to CFP ratio) values in cells arrested in G1 after Cdk4/6 inhibition (no Plk1 activity; 3.50 ± 0.50) in G2 cells treated with a Cdk1 inhibitor (no Plk1 activation; 3.79 ± 0.45) or in prometaphase cells treated with a Plk1 inhibitor (3.75 ± 0.45), but low FRET values (1.20 ± 0.45) in cells arrested in prometaphase with an Eg5[KIF11] inhibitor (fully active Plk1; Fig. 5A, B). Using these values as internal standards, we found that low doses of Aphidicolin decreased Plk1 activity in a dose-dependent manner in an interphasic cell population, which we know to be enriched in G2 cells[17] (200 nM: 1.91 ± 0.0.54, 400 nM: 2.35 ± 0.66, 600 nM: 3.65 ± 0.58, and 800 nM: 3.77 ± 0.57; Fig. 5A, B). This implied that mild replication stress (400 nM Aphidicolin) resulted in an intermediate (45% of maximal) Plk1 activity in G2. Consistent with our model, this intermediate Plk1 activity was suppressed by Cdk1 or Plk1 inhibition (Aph+Cdk1i: 3.39 ± 0.58; Aph+Plk1i: 3.53 ± 0.53; Fig. 5C, D). To validate this hypothesis, we further synchronised RPE1 cells expressing the Plk1-FRET sensor in G1 using CDK4/6 inhibitor and released them in the presence of DMSO, 200 nM Aphidicolin or 400 nM Aphidicolin. By quantifying Plk1 activity every 2 h, we found that mild replication stress had initially only had minor effects on Plk1 activity but that after 10 h, it prevented the full activation of Plk1, limiting its activity to intermediate levels (Fig. 5E, Supplementary Fig. S3A–D). We conclude that mild replication stress leads to an intermediate Plk1 activity in G2.

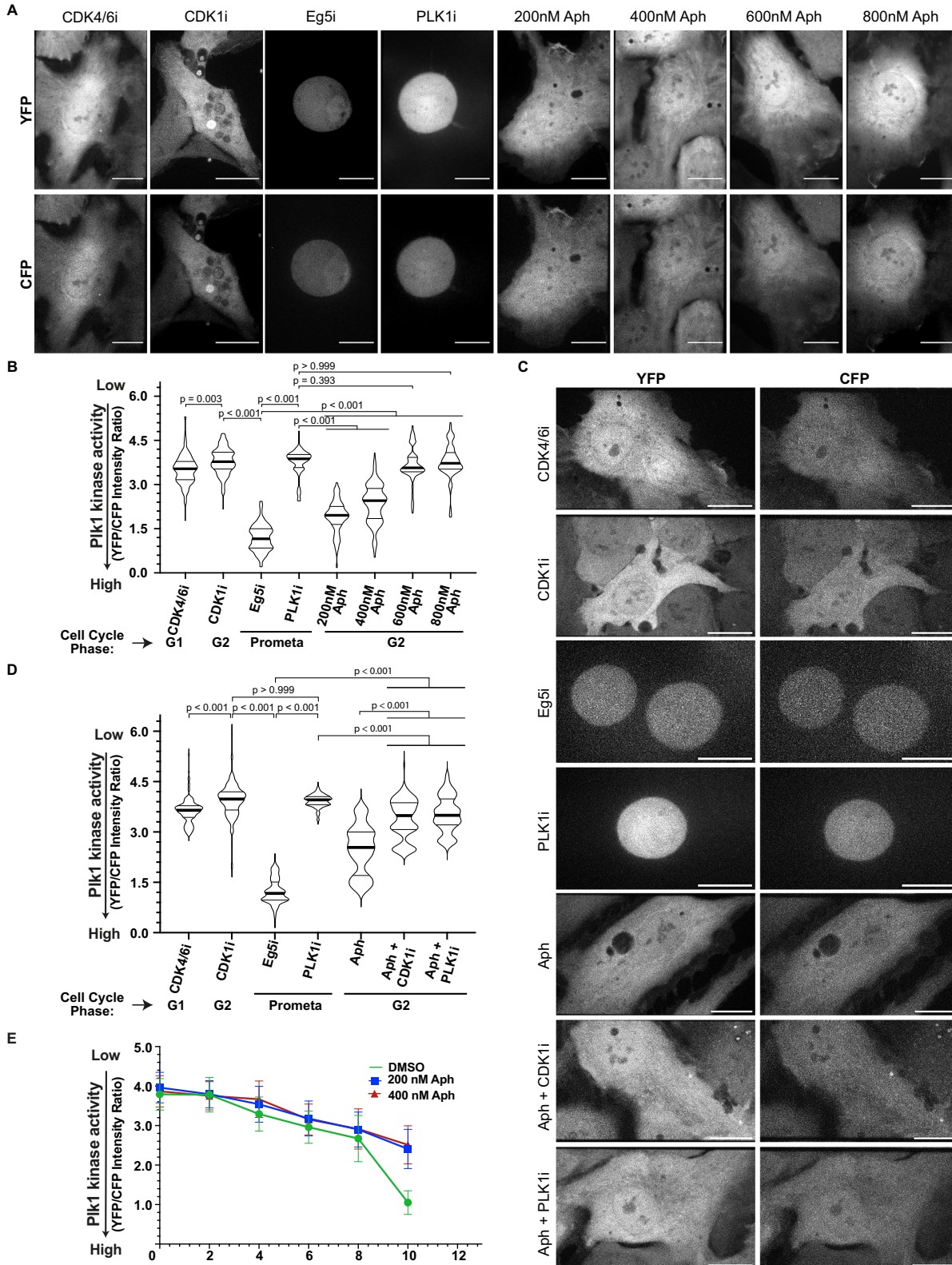

## ATR-Chk1 impose an intermediate Plk1 activity under mild replication stress

Activation of multiple players in the DDR pathway requires Plk1 activity, and Plk1 controls the recovery from G2/M arrest during DNA damage once the DNA repair is completed to facilitate mitotic entry[41–43]. To test whether the intermediate Plk1 activity is controlled by the ATR-Chk1 axis in Aphidicolin-treated cells, we again quantified

Plk1 activity using the FRET-based reporter. Inhibition of ATR or Chk1 kinase activity in Aphidicolin-treated cells fully rescued Plk1 activity as indicated by low FRET ratio values ($1.35 \pm 0.20$: ATRi, $1.29 \pm 0.09$: Chki vs. $2.28 \pm 0.66$ Aphidicolin alone; Fig. 6A–D). In contrast, ATM or Chk2 inhibition had no effect on Plk1 activity in Aphidicolin-treated cells ($2.50 \pm 0.29$: ATMi, $2.24 \pm 0.49$: Chk2i; Fig. 6A–D). This effect was specific for Aphidicolin-treated cells since ATR or Chk1 inhibition had

**Fig. 5 | Mild replication stress delays full Plk1 activation resulting in its sub-critical activity. A** Representative images of CFP and YFP fluorescence from RPE1 Plk1-FRET Sensor cells treated with indicated inhibitors/drugs. **B** Violin plots of YFP/CFP intensity ratios from cells in (**A**) [$N = 3$ independent experiments, $n = 162$ (Cdk4/6i), 164 (Cdk1i), 157 (Eg5i), 164 (Plk1i), 161 (200 nM Aph), 167 (400 nM Aph), 134 (600 nM Aph) and 132 (800 nM Aph) cells: $p$-values from Dunn's multiple comparison test]. **C** Representative images of CFP and YFP fluorescence from RPE1 Plk1-FRET Sensor cells treated with indicated inhibitors/drugs. **D** Violin plots of YFP/CFP intensity ratios from cells in C [$N = 3$ independent experiments, $n = 149$ (Cdk4/6i), 149 (Cdk1i), 142 (Eg5i), 154 (Plk1i), 148 (Aph), 135 (Aph+Cdk1i) and 143 (Aph+Plk1i); $p$ = Dunn's multiple comparison test]. Median in each case is marked

with a bold black line and thin grey lines denote the 1st and 3rd quartiles. **E** Quantification of dynamics of Plk1 activity in RPE1 cells as they progress through S and G2 phase after treatment with indicated drugs. Data plotted as mean ± SD ($N = 4$ independent experiments, $n = 210$ (0 h: DMSO), 198 (2 h: DMSO), 202 (4 h: DMSO), 183 (6 h: DMSO), 194 (8 h: DMSO), 177 (10 h: DMSO), 204 (0 h: 200 nM Aph), 199 (2 h: 200 nM Aph), 197 (4 h: 200 nM Aph), 191 (6 h: 200 nM Aph), 212 (8 h: 200 nM Aph), 200 (10 h: 200 nM Aph), 209 (0 h: 400 nM Aph), 197 (2 h: 400 nM Aph), 200 (4 h: 400 nM Aph), 185 (6 h: 400 nM Aph), 210 (8 h: 400 nM Aph) and 182 (10 h: 400 nM Aph); ($p$-values from two-tailed Sídak test)). Scale bars = 0.5 μm. Source data for all graphs are provided as a Source Data file.

no effect on Plk1 activity in prometaphase-arrested cells with an Eg5 inhibitor (1.20 ± 0.16: Eg5i+ATRi, 1.13 ± 0.27: Eg5i+Chk1i vs. 1.18 ± 0.25: Eg5i). We conclude that under mild replication stress conditions, the ATR-Chk1 axis but not the ATM-Chk2 axis prevents the full activation of Plk1, consistent with studies indicating that Chk1 acts upstream of Plk1 and that Chk1 can directly phosphorylate Plk1[44,45].

### Plk1 activates Separase and primes pericentrin to promote centriole disengagement

Plk1 induces centriole disengagement during mitotic exit by priming the pericentriolar protein Pericentrin for localised cleavage by the protease Separase[8–10]. To test whether Plk1 promotes premature centriole disengagement via the same molecular pathway, we studied the localisation of pericentrin in G2 cells with or without replication stress by super-resolution Stimulated emission depletion (STED) microscopy. While in DMSO-treated G2 cells, 96.40 ± 0.82% of the centrioles displayed a ~350 nm pericentrin ring, only 14.49 ± 4.10% of the centrioles in Aphidicolin-treated cells displayed such a complete ring (Fig. 7A, B, Supplementary Fig. S4A, B). An equivalent disruption of the Pericentrin ring was also found in control telophase cells (DMSO) when centrioles disengage normally (Fig. 7A and Supplementary Fig. S4C). A very similar loss of the pericentrin ring integrity could be observed after Aphidicolin treatment when using U-ExM (Fig. 7C, D). Finally, multi-colour U-ExM indicated that the disruption in Pericentrin ring integrity in cells could be rescued by inhibiting either Cdk1 or Plk1 (Fig. 7E, F) and that it correlated with the degree of centriole disengagement. (Fig. 7G). In contrast, the localisation of CEP57, another centriolar protein responsible for maintaining centriole engagement during mitosis[46], showed no change in localisation after Aphidicolin treatment (Fig. 7H, I).

Next, we quantified the contribution of Separase to premature centriole disengagement. We depleted Separase by siRNA (validation of depletion by immunoblotting see Supplementary Fig. S4D) in Aphidicolin or DMSO-treated cells (Fig. 8A) and monitored centriole disengagement. While Separase depletion had no effect on centriole disengagement in the absence of replication stress (Control siRNA: 3.80 ± 1.76%; Separase siRNA: 6.52 ± 4.70%; Fig. 8A, B), it strongly suppressed premature centriole disengagement in G2 in cells treated with Aphidicolin (14.62 ± 3.84% vs. 43.55 ± 3.20% in *siControl*; Fig. 8A, B). Moreover, Separase depletion restored in large parts the Pericentrin ring integrity around centrioles in Aphidicolin-treated cells G2 cells (76.38 ± 10.52% vs 27.68 ± 2.91% in *siControl*), while at the same time suppressing centriole disengagement (Fig. 8C–E). To exclude any off-target effects, we generated stable RPE1 cells expressing RNAi-resistant wild-type Myc-Separase under tetracycline-inducible promoter to test for a rescue of the disengagement phenotype (validation of exogenous Myc-Separase expression after control/Separase RNAi was confirmed by immunofluorescence; Supplementary Fig. S4E). We found that inducing Myc-Separase expression indeed rescued centriole disengagement as well as Pericentrin ring integrity disruption (Fig. 8F–H) under replication stress conditions. However, no centriole disengagement or disruption of the Pericentrin ring was observed after inducing Myc-Separase expression in the absence of Aphidicolin

(Fig. 8G, H). These results are consistent with a model in which Plk1 promotes premature centriole disengagement via a Separase-induced cleavage of the Pericentrin ring around centrioles (Fig. 9).

## Discussion

Here we investigated how mild replication stress in non-transformed cells disrupts the synchrony of the DNA and the centrosome cycle, leading to premature centriole disengagement. Using expansion microscopy, we demonstrate that mild replication stress induces premature centriole disengagement already in G2 via the ATR-Chk1-Wee1 axis of the DNA damage repair pathway. Activation of this pathway dampens but does not block the activity of the mitotic kinase Plk1, a critical regulator of both the DNA and the centrosome cycle. A sub-critical Plk1 activity is insufficient to promote rapid mitotic entry, resulting in a G2 delay; it is, however, sufficient to drive centriole disengagement via Separase and the disassembly of the Pericentrin ring, consistent with the canonical centriole disengagement pathway[8,47]. We thus propose that the differential threshold in Plk1 activity for mitotic entry and centriole disengagement is at the origin of the asynchrony in the DNA and centrosome cycle in cells experiencing mild replication stress.

In a previous study, we found that mild replication stress led to premature centriole disengagement and transient multipolar spindles in mitotic cells. When exactly centrioles became disengaged was nevertheless unknown due to the insufficient resolution of classical fluorescence microscopy. Using expansion microscopy, we now demonstrate that mild replication stress already causes centriole disengagement in G2 but not in the S phase. Given that ultrastructure expansion microscopy allows to study a high number of cells per experiment, this method has the potential to replace electron microscopy as the gold standard to determine whether centriole pairs are engaged or not. Our results further indicate that mild replication stress induces centriole disengagement via the ATR-Chk1-Wee1 pathway but not the ATM-Chk2 pathway. This indicates that under these conditions, disengagement is controlled by the single-strand break (SSB) arm of the DNA damage repair pathway. Whether activation of the ATM-Chk2 double-strand break arm of the DNA damage pathway also has the potential to create conditions favouring centriole disengagement remains to be seen.

The fact that centriole disengagement in G2 also depends on Cdk1/Cyclin A and Plk1, may at first appear counterintuitive since activation of the DNA damage repair pathway defers mitotic entry by preventing Cdk1 and Plk1 activation until DNA repair is completed[45,48]. Here we show that mild replication stress does not fully inhibit Plk1 activity; rather, our dynamic FRET measurements indicate that G2 cells with mild replication stress can partially activate Plk1 as cells progress through G2, reaching 50–60% of maximal activity, yet fail to reach the full activity seen just before mitotic entry. Since ATR inhibition alleviates this partial inhibition, we postulate that mild replication stress reduces Plk1 activity via the ATR-Chk1-Wee1 pathway to levels that are incompatible with a rapid mitotic entry yet sufficient to drive centriole disengagement. Our data are consistent with a model in which both processes require

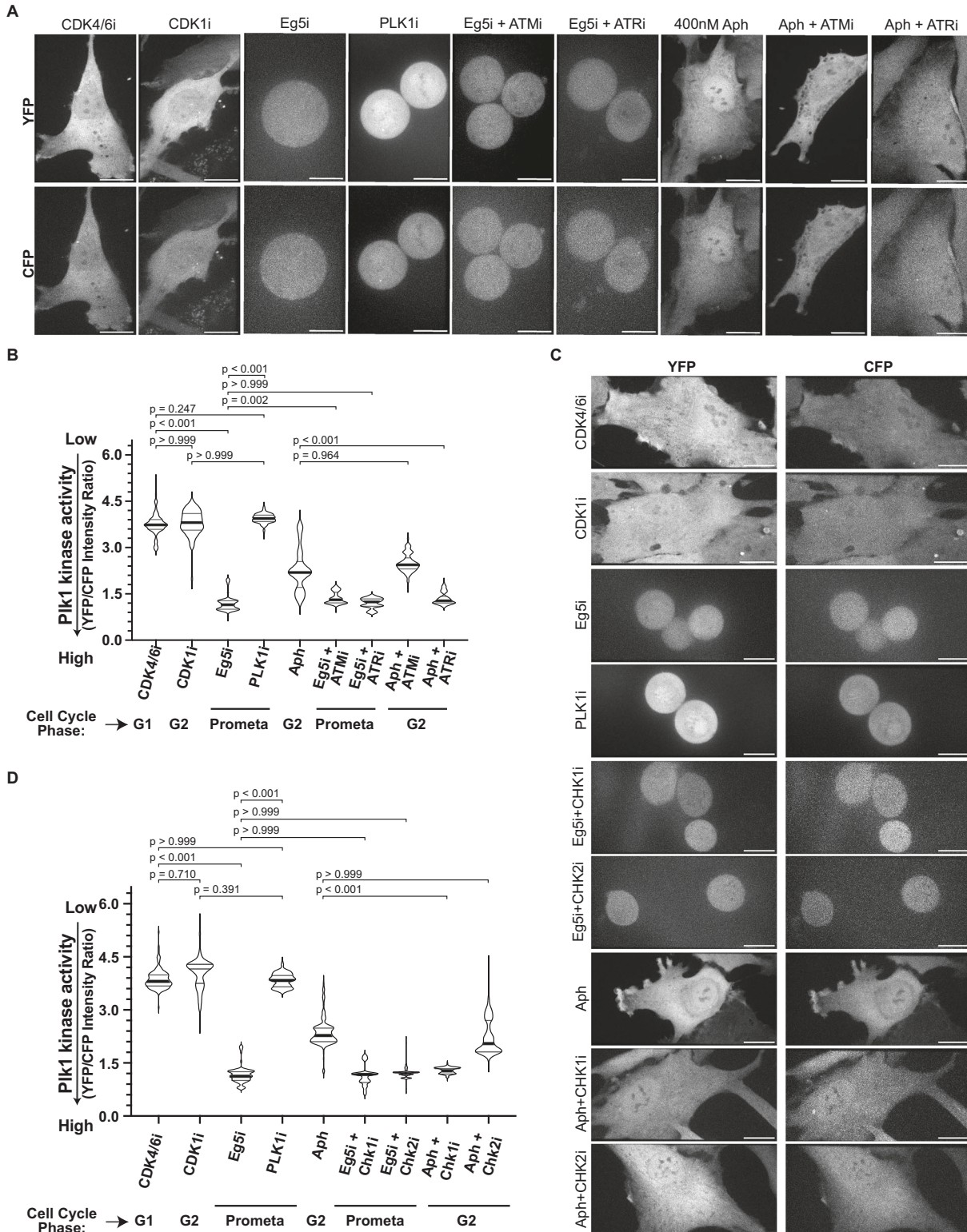

**Fig. 6 | ATR/Chk1 axis regulates the sub-critical Plk1 activity during replication stress. A** Representative images of CFP and YFP fluorescence from RPE1 Plk1-FRET Sensor cells treated with indicated inhibitors/drugs targeting indicated actors of the DNA damage repair pathway. **B** Violin plots of YFP/CFP intensity ratios from cells in A [*N* = 3 independent experiments, *n* = 146 (Cdk4/6i), 146 (Cdk1i), 146 (Eg5i), 151 (Plk1i), 149 (Aph), 151 (Eg5i+ATMi), 156 (Eg5i+ATRi), 159 (Aph+ATMi) and 159 (Aph+ATRi); *p*-values from Dunn's multiple comparison test]. **C** Representative images of CFP and YFP fluorescence from RPE1 Plk1-FRET Sensor cells treated with indicated inhibitors/drugs against DNA repair pathway proteins. **D** Violin plots of YFP/CFP intensity ratios from cells in C [*N* = 3 independent experiments, *n* = 152 (Cdk4/6i), 173 (Cdk1i), 178 (Eg5i), 175 (Plk1i), 176 (Aph), 180 (Eg5i+Chk1i), 181 (Eg5i+Chk2i), 176 (Aph+Chk1i) and 185 (Aph+Chk2i): *p*-values from Dunn's multiple comparison test]. Median in each case is marked with a bold black line and thin grey lines denote the 1st and 3rd quartiles. Scale bars = 0.5 μm. Source data for all graphs are provided as a Source Data file.

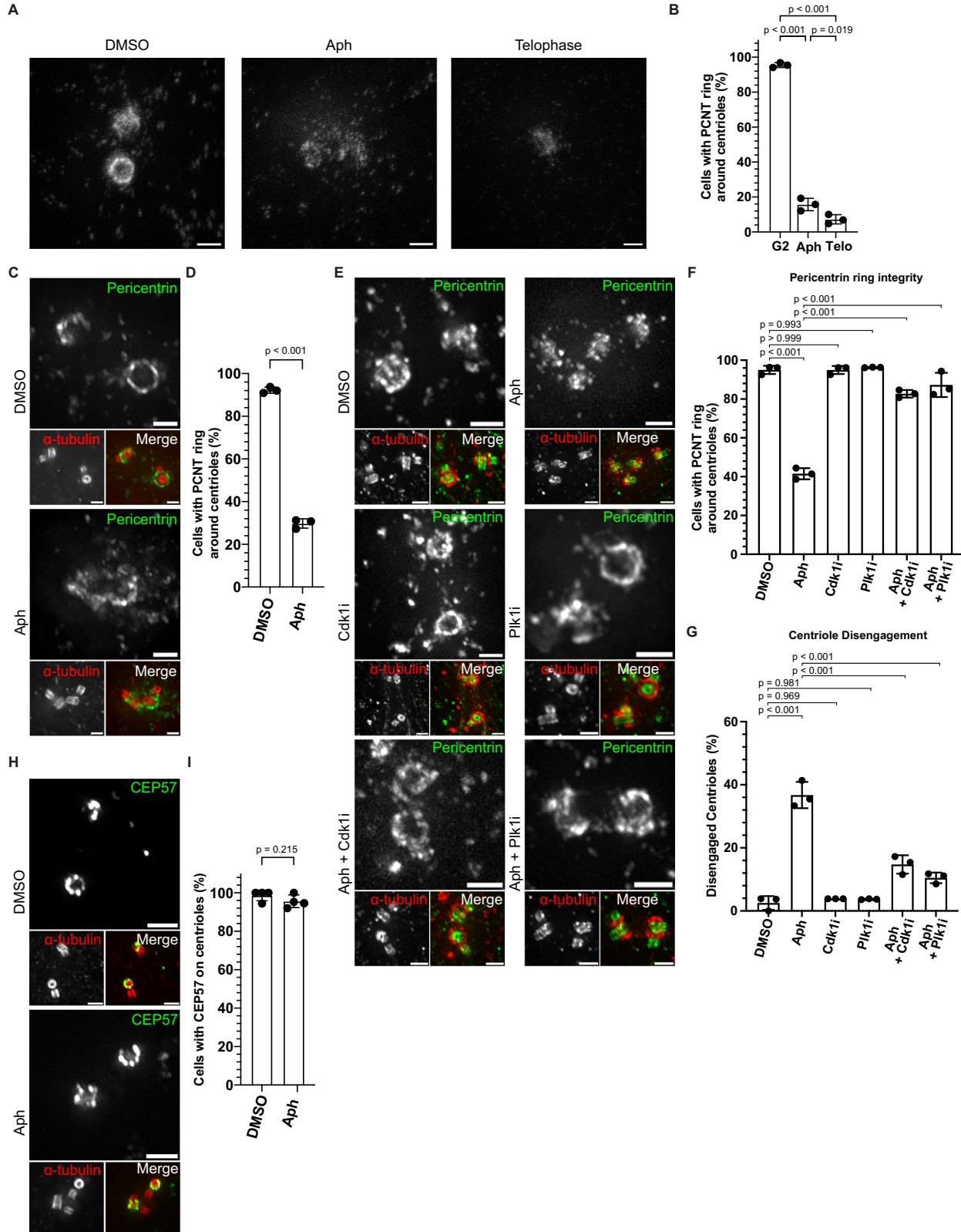

different Plk1 activity thresholds, explaining how mild replication stress can disrupt the synchronicity between the DNA and the centrosome cycle in late G2. Our model is also consistent with a previous study showing how severe DNA damage conditions that block mitotic entry can also induce a Plk1-dependent centriole disengagement in 40% of the cells after a long (48–72 h) G2 arrest[49]. We speculate the lower kinetics of centriole disengagement under

those conditions is due to lower residual Plk1 activity, consistent with the fact that the vast majority of the cells never enter mitosis.

If our hypothesis is correct, one key question is why a high Plk1 activity, which is present in unperturbed G2 cells, is not sufficient to drive centriole disengagement on its own. Under normal conditions, centriole disengagement occurs at the end of mitosis when Cdk1/ Cyclin-B1 has been inactivated by APC$^{Cdc20}$ mediated protein

**Fig. 7 | Mild replication stress affects PCM integrity. A** STED nanoscopy images of zoomed pericentriolar region of G2 phase RPE1 cells treated either with DMSO or Aph and DMSO-treated cells in telophase. **B** Quantification for the percentage of G2 cells having intact pericentrin as ring in conditions depicted in (**A**) ($N = 3$ independent experiments, $n = 71$, 78 and 70 for DMSO: G2 phase, Aph and DMSO: Telophase cells, respectively: $p$-values from two-tailed Sídak test). **C** U-ExM images of centrioles in G2 Phase RPE1 cells treated either with DMSO or Aph and stained for α-tubulin (red) and Pericentrin (green). **D** Quantification of the percentage of G2 phase cells with complete Pericentrin ring around centrioles depicted in C ($N = 4$ independent experiments, $n = 82$ and 76 cells in DMSO and Aph, respectively: $p$-values from two-tailed $t$-test). **E** U-ExM images of centrioles in G2 Phase hTERT-RPE1 cells treated either with indicated drugs and stained for α-tubulin (red) and Pericentrin (green). **F** Quantification of G2 phase cells with complete Pericentrin ring around centrioles depicted in E ($N = 3$ independent experiments, $n = 80$: DMSO, 87: Aph, 78: RO3306, 81: BI2536, 81: Aph+RO3306 and 96: Aph+BI2536: $p$-values from two-tailed Sídak test). **G** Quantification of G2 phase cells with disengaged centrioles ($N = 3$ independent experiments, $n = 80$: DMSO, 87: Aph, 78: RO3306, 81: BI2536, 81: Aph+RO3306 and 96: Aph+BI2536: $p$-values from two-tailed Sídak test). **H** U-ExM images of centrioles in G2 Phase RPE1 cells treated either with indicated drugs and stained for α-tubulin (red) and CEP57 (green). **I** Quantification of the percentage of G2 phase cells with complete CEP57 ring around their centrioles ($N = 4$ independent experiments, $n = 65$ and 83 cells in DMSO and Aph, respectively: $p$-values from two-tailed $t$-test). Data presented as mean values ± SD. Scale bars = 0.5 μm. Source data for all graphs are provided as a Source Data file.

degradation[50] and Plk1 activity started to drop due to APC/C$^{Cdh1}$ mediated degradation[51]. This could point to an inhibitory role of Cdk1/CyclinB1 during late G2 and mitosis that prevents a high Plk1 activity from inducing premature centriole disengagement. This could be achieved by the near simultaneous activation of both kinases during mitotic entry, which under normal conditions, prevents an exclusive activation of Plk1. One plausible target of Cdk1/Cyclin B1 could be Separase, which we identify as a key regulatory of premature centriole disengagement and which is inhibited by Cdk1/CyclinB1 activity[52]. The synchronous loss of Securin and Cyclin-B1 at anaphase onset by APC/C$^{Cdc20}$ could thus not only activate Separase to induce chromosome segregation[53] but also permit centriole disengagement under the control of Plk1. A second important question is how Separase can act on centrosomes in G2 in Aphidicolin-treated cells without affecting chromosome cohesion. We reason that one main reason is that Separase is excluded from the nucleus before mitotic entry by CRM1-dependent nuclear export to prevent cohesin cleavage during chromosome condensation[54]. Furthermore, cytosolic Separase is thought to be inactivated by Securin in addition to Cdk1 phosphorylation at S1126[55,56]. Our results indicate that in the absence of robust Cdk1/Cyclin B1 activity, Securin inhibition is not sufficient, resulting in premature centriole premature centriole disengagement.

Downstream of Separase and Plk1, we identify loss of the structural integrity of the Pericentrin ring and, more generally, the PCM as likely pre-requirement for premature centriole disengagement in cells experiencing mild replication stress. This would indicate that premature centriole disengagement in G2 depends in part on the canonical centriole disengagement pathway in which Separase-induced cleavage at R2231 initiates Pericentrin degradation during mitotic exit[9]. We did, however, not observe any change in Cep57 localisation, pointing to potential differences between the two processes. Whether premature centriole disengagement purely relies on Pericentrin cleavage or whether other components are involved remains an open question.

We conclude that replication stress in non-cancerous cells deregulates the synchrony between the cell and centriole duplication cycle and thus connects the DNA damage response pathway and the centriole disengagement via Plk1. This deregulation is much more pernicious than the premature centriole disengagement seen after strong DNA damage[49,57] since cells experiencing only mild replication stress will nevertheless enter mitosis, forming transient multipolar spindles. Replication stress and deregulation of the centrosome cycle are both hallmarks of pre-cancerous and cancerous lesions[58–60]. To which extent the two phenomena are linked is, however, not clear. Our study uncovers a new connection between both processes, providing a possible mechanistic explanation for their co-evolution in cancer cells. Multiple theories do exist pertaining to the origin of centrosome cycle deregulation and appearance of supernumerary centrosomes in cancer cells, including cytokinesis failure, uncontrolled centriole duplication or elongation and PCM disintegration during mitosis[61,62]. Here, we speculate that a lack of synchrony between the centrosome and DNA cycle may also favour the appearance of abnormal centrosome number in cells experiencing persistent replication stress, as cells might accumulate disengaged centrioles that are prematurely licensed for centrosome duplication.

## Methods

### Cell culture and drug treatments

hTERT-RPE1 (ATCC: CRL-4000) and hTERT-RPE1 EB3-eGFP/H2B-mCherry cells (kind gift by Willy Krek, ETH Zurich, Switzerland) were cultured in Dulbecco's Modified Eagle's Medium (Thermo Fisher Scientific: 61965-026) supplemented with 10% fetal calf serum (FCS) (Biowest: S181S-500) and 100 U/ml of each penicillin and streptomycin (P/S) (Thermo Fisher Scientific: 15140122) at 37 °C, 95% relative humidity and 5% CO$_2$ in humidified CO$_2$ incubator. hTERT-RPE1 FRT/TR (kind gift from Johnathan Pines, Institute of Cancer Research, United Kingdom), hTERT-RPE1 OSTR1 cells, hTERT-RPE1 Cyclin-A2-double degron cells and hTERT-RPE1 Cyclin-B1-double degron cells (all kind gifts by Helfrid Hochegger, University of Sussex, United Kingdom) were cultured in Dulbecco's Modified Eagle's Medium (Thermo Fisher Scientific: 61965-026) supplemented with 10% tetracycline free fetal calf serum (FCS) (Biowest: S181T-500) and 100 U/ml of each penicillin and streptomycin (P/S) (Thermo Fisher Scientific: 15140122) at 37 °C, 95% relative humidity and 5% CO$_2$ in humidified CO$_2$ incubator. hTERT-RPE1 Plk1 FRET Sensor cells were prepared by transfecting Plk1-FRET sensor c-jun substrate plasmid[40] (Addgene: 45203) in hTERT-RPE1 cells using X-tremeGENE™ 9 (Merck: XTG9-RO) transfection reagent according to manufacturer's instructions. Transfected cells were selected with 600 μg/ml of G418 (Invivogen: ant-gn-5) in DMEM with 10% FCS and 100 U/ml P/S, followed by single cell cloning. hTERT-RPE1: Myc-Separase (RNAi-resistant) cells were prepared by transfecting hTERT-RPE1 FRT/TR cells with 0.5 μg Myc-Separase (WT) and 4.5 μg pOG44 plasmids using X-tremeGENE™ 9 (Merck: XTG9-RO) transfection reagent according to manufacturer's instructions. The transfected recombinants were selected with 200 μg/ml of 200 μg/ml Hygromycin-B (Invivogen: ant-hm-5) in DMEM with 10% FCS and 100 U/ml P/S. The plasmid for RNAi-resistant Myc-Separase (WT) was prepared by site-directed mutagenesis to introduce silent mutations in Separase open reading frame using QuickChange II site-directed mutagenesis kit (Agilent: 200521) using pcDNA5-FRT/TO-myc-hSeparase (Addgene: 59820)[63] as template. All cell lines were routinely tested for mycoplasma contamination by PCR. For live-cell imaging, cells were cultured in Leibovitz's L-15 medium without Phenol Red (Thermo Fisher Scientific: 21083-027) with 10% FCS and 100 U/ml P/S at 37 °C.

To induce mild replication stress in all cell types, Aphidicolin (Sigma Aldrich: A0781) was dissolved in DMSO and applied at 400 nM for 16 h. For auxin-induced degradation of cyclins, cells were treated with 1 μg/ml Doxycycline (Sigma Aldrich: D9891-1G) for 2 h, followed by the addition of 3 μM Asunaprevir (Apexio: BMS-650032) and 500 μM 3-Indoleacetic acid (Sigma Aldrich: I2886). Cells were treated with following inhibitors to inhibit indicated cellular proteins, Eg5: 5 μM STLC (Sigma Aldrich: 164739), Cdk4/6:

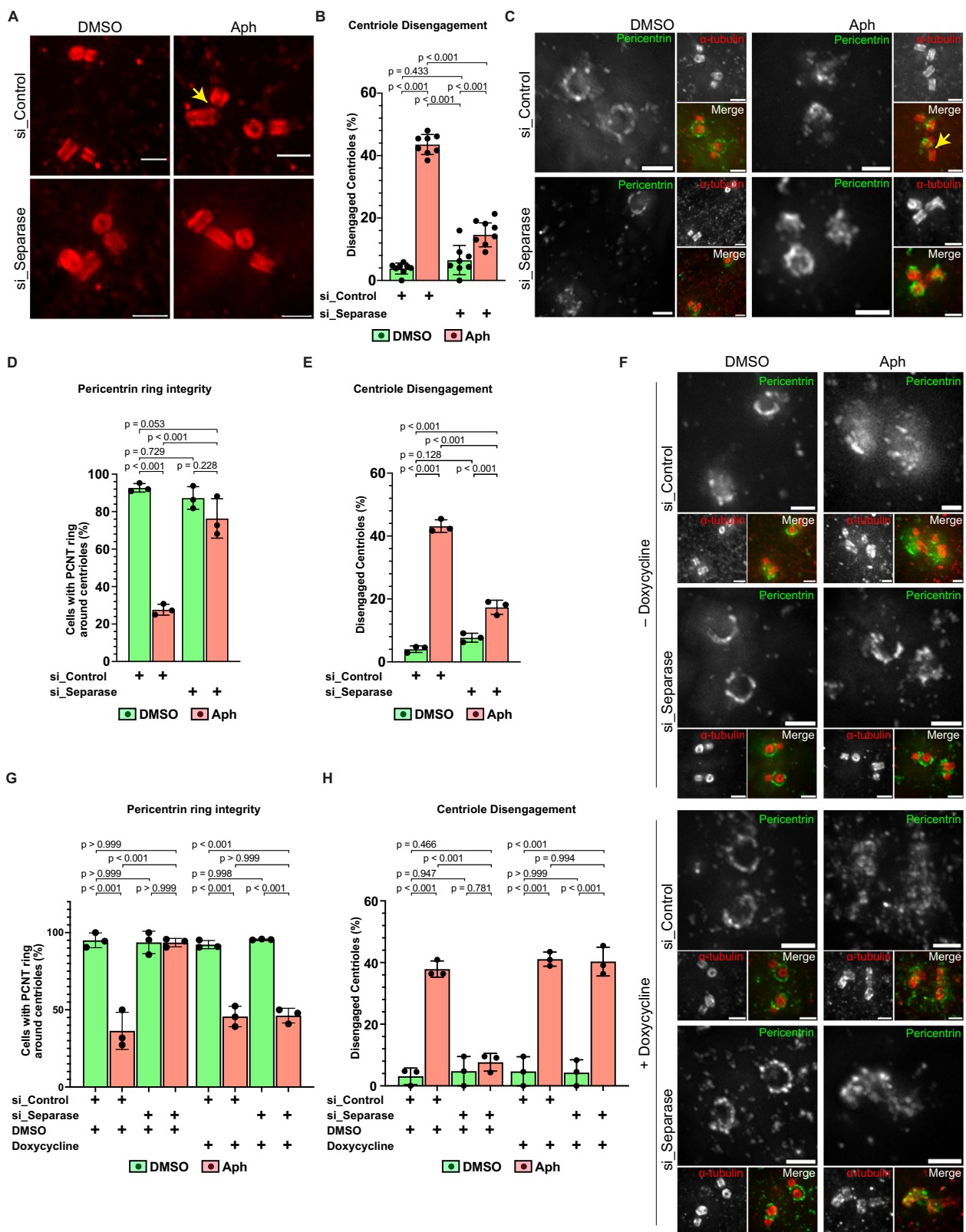

500 nM Palbociclib/PD0332991 (Cayman Chemical Company: 16273), Cdk1: 9 μM RO3306 (Sigma Aldrich: SML0569), Plk1: 10 nM BI2536 (Selleck Chemicals: S1109), ATR: 0.8 μM ETP-46464 (Apexbio: A8626), ATM: 10 μM KU-55933 (Selleck Chemicals: S1092), Chk1: 5 μM LY2603618 (Chk1i-1) (Selleck Chemicals: S2626), Chk1: 1 μM PF-477736 (Selleck Chemicals: S2904), Chk2: 10 μM BML-277 (Selleck Chemicals: S8632), Wee1: 0.5 μM PD0166285 (Wee1i-1) (Selleck

Chemicals: S8148), and Wee1: 0.5 μM MK-1775 (Selleck Chemicals: S1525). For synchronisation, cells were treated with 500 nM of Palbociclib for 24 h and released in a complete medium.

## RNA interference

siRNA transfections were performed using Lipofectamine RNAiMAX (Thermo Fisher Scientific: 13778075) according to the manufacturer's

**Fig. 8 | Premature centriole disengagement requires Separase-dependent PCM disintegration. A** U-ExM images of centrioles in G2 phase RPE1 cells treated either with siControl or siSeparase and DMSO or Aph. **B** Quantification of G2 phase cells having engaged centrioles depicted in (**A**) (*N* = 8 independent experiments, *n* = 187, 195, 188 and 192 cells in siControl:DMSO, siControl:Aph, siSeparase:DMSO and siSeparase:Aph, respectively: *p*-values of two-tailed Sídak test). **C** U-ExM images of centrioles in G2 phase RPE1 cells treated either with DMSO or Aph and siControl or siSeparase and stained with α-tubulin (red) and Pericentrin (PCM: green). **D** Quantification of G2 phase cells with complete Pericentrin ring around their centrioles depicted in **C** (*N* = 3 independent experiments, *n* = 68, 70, 60 and 61 cells treated with siControl:DMSO, siControl:Aph, siSeparase:DMSO and siSeparase:Aph, respectively: *p*-values from two-tailed Sídak test). **E** Quantification of G2 phase cells with disengaged centrioles depicted in **C** (*N* = 3 independent experiments, *n* = 68, 70, 60 and 61 cells treated with siControl:DMSO, siControl:Aph, siSeparase:DMSO and siSeparase:Aph, respectively; *p*-values from two-tailed Sídak

test). **F** U-ExM images of centrioles in G2 phase RPE1:Myc-Separase cells treated with indicated inhibitors/drugs along with either siControl or siSeparase and stained with α-tubulin (red) and Pericentrin (green). **G** Quantification of G2 phase cells with complete Pericentrin ring around centrioles depicted in F (*N* = 3 independent experiments, *n* = [64: si_Control+DMSO, 66: si_Control+Aph, 64: si_Separase+DMSO, 66: si_Separase+Aph] and [64: si_Control+DMSO, 66: si_Control +Aph, 67: si_Separase+DMSO, 67: si_Separase+Aph: *p*-values from two-tailed Sídak test] in absence or presence of doxycycline, respectively). **H** Quantification of G2 phase cells with disengaged centrioles depicted in F (*N* = 3 independent experiments, *n* = [64: si_Control+DMSO, 66: si_Control+Aph, 64: si_Separase+DMSO, 66: si_Separase+Aph] and [64: si_Control+DMSO, 66: si_Control+Aph, 67: si_Separase +DMSO, 67: si_Separase+Aph] in absence or presence of doxycycline, respectively: *p*-values from two-tailed Sídak test). Data presented as mean values ± SD. Scale bars = 0.5 μm. Source data for all graphs are provided as a Source Data file.

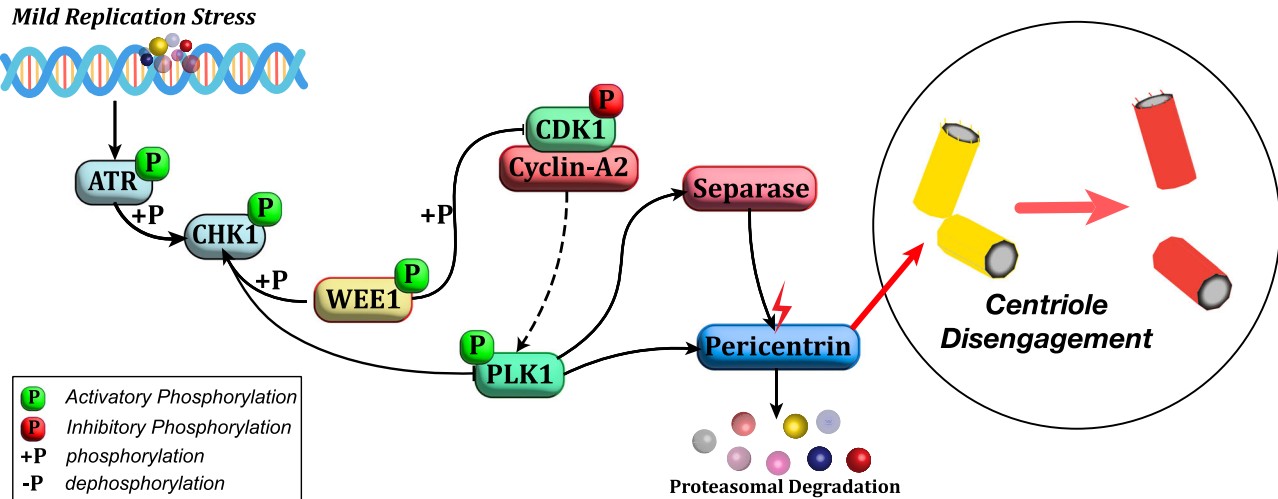

**Fig. 9 | Schematic representation of the molecular pathway regulating premature centriole disengagement during mild replication stress.** Postulated model of how mild Replication stress causes activation of the ATR-Chk1 axis of DNA

damage repair pathway, resulting in a sub-critical Plk1 activity that favours separase-dependent premature centriole disengagement in G2.

instructions. RNAi was performed for 72 h; when combined with inhibitors, the drugs were added 60 h post-transfection. All the siRNA sequences used in the study were previously validated sequences: siControl (Qiagen, GGACCTGGAGGTCTGCTGT) and siSeparase (Dharmacon, GCTTGTGATGCCATCCTGA)[64].

## Antibodies
The following antibodies were used in this study: Mouse anti-α-tubulin (Geneva antibody facility: AA345-M2a; 1:250: ExM)[65], Mouse anti-β-tubulin (Geneva antibody facility: AA344-M2a; 1:250: ExM)[65], Mouse anti-α-tubulin (Clone: DM1α, Sigma Aldrich, T9026, 1:5000: Western Blotting), Rabbit polyclonal anti-Pericentrin (abcam: ab4448; 1:250: ExM, 1:1000: STED; 1:2000: Immunofluorescence), Rabbit polyclonal anti-CEP57 (GeneTex: GTX115931; 1:250: ExM), Mouse anti-Cyclin-A2 (Clone: E32.1, abcam: ab38; 1:1000: Western Blotting), Mouse anti-Cyclin-B1 (Clone: Y106, abcam: ab72; 1:1000: Western Blotting), Mouse anti-Separase (abcam: ab16170; 1:1000: Western Blotting), Mouse anti-c-Myc (Thermo Scientific: MA1-980; 1:1000: Immunofluorescence) and Mouse anti-CENP-F (abcam: ab90; 1:1000: STED). All the Alexa Fluor-conjugated secondary antibodies were purchased from Thermo Fisher Scientific and used at 1:500 dilution. HRP-conjugated goat anti-mouse antibody for western blotting was purchased from Thermo Fisher (Cat # 32430) and used at 1:10,000 dilution. Goat anti-rabbit STAR RED antibody for STED microscopy, purchased from Abberior Instruments GmbH (Cat # 2-0002-011-9), was used at 1:1000 dilution.

## Expansion microscopy
Cells were grown on 12 mm circular glass coverslips (Thermo Fisher Scientific) and treated with required inhibitors/drugs overnight. Next day the coverslips were treated with Acrylamide (AA)-Formaldehyde (FA) Solution [1.4% AA (Sigma Aldrich: A4058) and 2% FA (Sigma Aldrich: F8775) in PBS] for 5 h at 37 °C to prevent protein crosslinking. Coverslips were next subjected to gelation by incubation for 1 h at 37 °C with monomer solution [19% Sodium Acrylate (Sigma Aldrich: 408220), 10% Acrylamide, 0.1% Bisacrylamide (Sigma Aldrich: M1533), 0.5% Tetramethyl ethylenediamine-TEMED (Thermo Fisher: 17919), 0.5% Ammonium Persulfate (Thermo Fisher: 17874) in PBS]. Post gelation, the coverslips were treated with denaturation solution [50 mM Tris (Sigma Aldrich: 99362), 200 mM Sodium dodecyl sulphate (Axon Lab AG: A2572.0500), 200 mM Sodium Chloride (Axon Lab AG: A3597.1000) in Nuclease free water, pH: 9.0] for 15 min on a rocker shaker at room temperature to detach the gels from coverslips. The gels were heated at 95 °C for 90 min in a denaturation solution followed by three 30 min washes with water. The gels were incubated with PBS for 15 min followed by 3 h incubations with primary and secondary antibodies followed each by three 10 min washes at 37 °C and gentle shaking. Stained gels were kept overnight in water for optimal expansion. The size of the gel was measured to calculate the expansion factor, and the gel was cut into small pieces and placed in 2 well plastic bottom ibidi chamber (Ibidi GMBH: Cat # 80286). The 3D image stacks of centrioles in G2 phase cells (4 centrioles in one cell) were acquired in 0.1 μm steps using a 100x oil-immersion (NA 1.4)

objective on an Olympus DeltaVision microscope (GE Healthcare) equipped with a DAPI/FITC/Rhodamine/CY5 filter set (Chroma Technology Corp) and a CoolSNAP HQ camera (Roper-Scientific). The three-dimensional image stacks were deconvolved with SoftWorx (GE Healthcare). The acquired images were cropped and processed with ImageJ (NIH) software to construct 3D images to analyse the configuration of centrioles (orthogonal orientation and distance in between) for each image.

### Live-cell imaging and analysis

For live-cell imaging experiments, hTERT-RPE1 cells stably expressing EB3-eGFP and H2B-mCherry were plated in glass bottom Ibidi chambers (Ibidi GMBH: Cat # 81158), and normal DMEM medium was replaced with L15 Leibovitz's medium supplemented with 10% FCS and 100 U/ml P/S. The cells were treated with indicated inhibitors and imaged at 37 °C on a Nikon Ti microscope equipped with a 60x NA 1.3 oil objective, DAPI/FITC/Rhodamine/CY5 (Chroma, USA) filter set, Orca Flash 4.0 CMOS camera (Hamamatsu, Japan) and the NIS software. Cells were recorded every 3 min for 18 h with z-slices separated by 2 μm and 100 ms exposure per z-slice at wavelengths of 488 (525) and 561 nm (615 nm) excitation (emission). The time-lapse movies were analysed manually for multipolarity using Imaris software (Bitplane Inc).

### STED nanoscopy

Cells were grown on glass coverslips (170 ± 10 μm thick Hecht-Assistent: Cat # 41014515) and fixed with chilled methanol at −20 °C for 6 min. Coverslips were stained with Rabbit anti-Pericentrin and Mouse anti-CENP-F (to identify G2 phase cells). Coverslips with fluorescent labelled samples were mounted on microscope slides using Fluoromount-G™ (Thermo Fisher Scientific: Cat # 00-4958-02) and sealed with nail paint from all sides. Dual colour 2D-STED imaging was performed on a TCS SP8 STED microscope (Leica Microsystems) at 21 °C using a STED motorised oil-immersion objective (HCPL Apo 100×/NA 1.30 motCOR) using LAS-X Imaging software (Leica Microsystems). Excitation was performed with a white light laser (WLL) and depletion with a 775-nm pulsed laser. Both the excitation and depletion lasers were calibrated either with the STED Expert Alignment Mode and Abberior gold nanoparticles (diameter: 80 nm) before starting each imaging session or with the STED Auto Beam Alignment tool during imaging sessions (Leica LAS X software). The STED imaging was made sequentially using excitation at 580 nm (WLL) and an STED 775 depletion laser line for Abberior STAR RED (for Pericentrin) anti-rabbit antibody. Detection signals were collected between 647 and 677 nm for STAR RED using highly sensitive Leica Hybrid Detectors with a fixed gain and offset (100 mV and 0, respectively). Time-gated detection was used for all fluorophores (0.50–6.00 ns). Acquisitions were performed with a line average of 4, a speed of 400 Hz, and software optimised pixel size respecting the Nyquist criteria. 2D-STED images were deconvolved using the Leica Lightning Mode (LAS X software), and the analysis for Pericentrin structure was performed with ImageJ (National Institutes of Health). Cells were co-stained with CENP-F in the AF488 channel, and the signal was used only to identify late G2 phase cells.

### FRET assay to measure Plk1 activity

hTERT-RPE1 Cells stably expressing Plk1 FRET sensor were seeded in four-well glass bottom μ-Slide ibidi chambers (Ibidi; 80426) and treated with different inhibitors/drugs as indicated in DMEM with 10% FCS and 1% P/S. The DMEM was replaced with Leibovitz L-15 supplemented with 10% FCS and 1% P/S containing the same drugs/inhibitors as before. The chambers were acclimatised in a 37 °C chamber before imaging. The acquisition was performed with an EC Plan Neofluor 100X (NA 1.3) oil objective on a Zeiss Cell Observer.Z1 spinning disk microscope (Nipkow Disk) equipped with a 37 °C chamber and a CSU X1 automatic Yokogowa spinning disk head. To perform FRET

experiments, samples were illuminated with a 445 nm laser and the emission signal was split equally using a DV2 split view system and CFP and YFP emissions were recorded on the split beams. Then, 512×512 pixel size images were acquired with an Evolve EM512 camera (Photometrics) using Visiview 4.00.10 software. Acquired images were analysed using ImageJ to calculate the YFP to CFP emission intensity ratio after background subtraction.

### Immunofluorescence

hTERT-RPE1 Cells were grown on acid-etched 24×24 mm glass coverslips (Huberlab AG: 10.0360.10) and treated with drugs and siRNAs as indicated in DMEM with 10% FCS and 1% P/S. The cells were fixed in ice-cold methanol. After fixation, the coverslips containing cells were rinsed with PBS and blocked for 30 min in PBS+3% BSA (PAN-Biotech; P06-1391500), followed by incubation with primary and secondary antibody solutions for 1 h and 30 min, respectively. All the primary and secondary antibodies were diluted in PBS+3% BSA. The cells, after immunolabelling, were washed thrice with PBS before mounting the coverslips with VECTASHIELD with DAPI (Vector Laboratories; H-1200, H-1000). 3D immunofluorescence images were acquired in 0.2 μm z-steps using a 60x oil-immersion (NA 1.4) objective on an Olympus DeltaVision microscope (GE Healthcare) equipped with a DAPI/FITC/Rhodamine/CY5 filter set (Chroma Technology Corp) and a CoolSNAP HQ camera (Roper-Scientific). The three-dimensional image stacks were deconvolved with SoftWorx (GE Healthcare). The acquired images were cropped and processed with ImageJ (NIH) software.

### Immunoblotting

Cells were grown in 60 mm plastic dishes and treated with inhibitors/drugs overnight. To prepare protein lysate, the cells were scrapped off using a cell scrapper and lysed in RIPA buffer (50 mM Tris pH-7.4, 150 mM NaCl 1% Nonidet P-40 (Thermo Fisher Scientific: 85124), 0.5% Sodium deoxycholate (Sigma Aldrich: D5670), 0.1% Sodium dodecyl sulphate in ultrapure water) supplemented with Protease inhibitor (Roche: 11873580001) and Phospho-STOP (Roche: 04906845001). Protein concentrations in the lysates were determined using the Bradford Protein Assay (Thermo Fisher; 23200). Samples with equal amounts of protein were mixed with 5X Laemmli buffer and heated to 95 °C for 5 min. Proteins were separated on a 10% SDS-polyacrylamide gels and transferred onto a 0.45 μm pore size nitrocellulose membrane (Macherey-Nagel GMBH: 741280) by wet blotting. Membranes were blocked with 5% non-fat dry milk in PBS 0.2% Tween20 (PBS-T) for 30 min. After blocking, membranes were incubated with primary antibodies overnight at 4 °C in PBS-T 5% non-fat dry milk. Membranes were washed three times with PBS-T and incubated for 1 h with the appropriate peroxidase-conjugated secondary antibody in PBS-T 5% non-fat dry milk. The membranes were washed thrice with PBS-T, and the bands corresponding to protein of interest were detected by chemiluminescence using the Amersham ECL Prime Western Blotting Detection Kit (GE Healthcare; RPN2232) in a Fusion FX7 Spectra Multispectral Imaging system (Witec AG, Switzerland).

### Statistical analysis

Statistical tests for all figures were performed using GraphPad Prism 9 (GraphPad); the statistical tests employed in every case are described in the figure legends. A minimum of three independent biological replicates were performed in all experiments.

### Reporting summary

Further information on research design is available in the Nature Portfolio Reporting Summary linked to this article.

## Data availability

All the raw data related to figures and supplementary figures (representative images and movies) are available at: https://doi.org/10.

26037/yareta:4f2f6xydgjfmvc6alqq6fbv2aq. Due to the large size (>4TB), the raw live-cell imaging data used for analyses will be made available on request by sending external hard disks. Source data are provided with this paper.

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

## Acknowledgements

The authors thank H. Hochegger (University of Sussex, UK) and J. Pines (Institute of Cancer Research, UK) for cell lines, P. Guichard, M. Laporte and V. Hamel (University of Geneva, Switzerland) for expansion microscopy support, Members of Bioimaging and FACS facility (University of Geneva, Switzerland) for experimental support, Monica Gotta and team (University of Geneva, Switzerland) and members of Meraldi laboratory for helpful discussions and support, and P. Guichard, V. Hamel and M. Gotta for critical comments on the manuscript. This work in the Meraldi laboratory was supported by the Swiss National Science Foundation (Schweizerischer Nationalfonds zur Förderung der Wissenschaftlichen Forschung; SNF) project grant (No. 31003A_179413 & 310030_208052) and the Université de Genève.

## Author contributions

Conceptualisation: D.D., P.M.; Formal analysis: D.D., P.M.; Investigation: D.D., D.H.; Writing—original draft: D.D.; Writing—review & editing: D.D., P.M.; Visualisation: D.D.; Supervision: P.M.; Project administration: P.M.; Funding acquisition: P.M.

## Competing interests

The authors declare no competing interests.
