## [Peer Review File · Nature Communications]

Mild replication stress causes premature centriole disengagement via a sub-critical Plk1 activity under the control of ATR-Chk1REVIEWER COMMENTS

Reviewer #1 (Remarks to the Author):

In the present manuscript, Dwivedi et al. provide further mechanistic insights into how mild replication stress causes premature centriole disengagement. The present manuscript relies on a previous publication by the Meraldi group where a clear link between mild replication stress, premature centriole disengagement and chromosome segregation errors was reported (Wilhelm et al., Nat Commun, 2019). Cdk1 and Plk1 requirement upstream of centriole disengagement have already been described in the previous manuscript. The main novelty of the manuscript by Dwivedi is the notion of a sub-critical level of Plk1 activity during the G2 phase of the cell cycle, unravelled by an elegant use of a Plk1 FRET biosensor. Restrained Plk1 activity level in G2 appears not sufficient for premature mitotic entry but sufficient to prime unscheduled separase-dependant PCM disassembly according to the authors. If this mechanistic explanation appears appealing, the authors need to provide more experimental evidences and clarify important points to fully support their model before the manuscript is suitable for publication. The manuscript is well written, referenced and presented, even if some clarifications are required.

Here are my major concerns on the manuscript:

1) In the previous publication, premature centriole disengagement was noticed in early mitosis after conventional microscopy. In this manuscript, the authors observe premature centriole disengagement in G2 phase of the cell cycle after expansion microscopy technique. In Figure 1, the examples provided do not enable to clearly distinguish between normal and abnormal centriole configurations. For example, to my sense, centriole configurations appear similar in Fig1B and Fig1D after aphidicolin treatment, contrary to what is quantified in Fig1C and Fig1E, where 100% of centrioles appear engaged in S phase but only 55% in G2 phase. What shall we believe? Is it a misleading choice of images?

The fact that centriole disengagement does not occur before G2 phase is important to support the model proposed by the authors and more experiments are required to show that premature centriole disengagement is restricted to G2. The authors could synchronize the cells in G1 with CDK4/6 inhibitor before release in S phase with low dose aphidicolin treatment and analysis of centriole configurations in G2 phase (at least 8h post-release?).

2) In the great majority of examples provided, only one of the two centrosomes display premature centriole disengagement (one arrow in Fig1D, 2B, 3D...). How can the authors explain this asymmetry? Can they quantify it? Is the mother or daughter centrosome more prone to centriole disengagement? If this is the case, pericentrin disassembly driven by separase activity and responsible for centriole disengagement according to the authors should also be asymmetric. Nonetheless, the authors observe

pericentrin disassembly on both centrosomes according to the images provided in Fig7, 8 and S3. How can the authors explain this apparent discrepancy?

3) The authors used a FRET biosensor to unravel a sub-critical level of Plk1 activity in G2 after low doses of aphidicolin. The notion of sub-critical level relies on a direct comparison of Plk1 activity level with G1 phase (lower level) and prometaphase (higher level). The authors provide only one value of Plk1 activity for the whole G2 phase without considering the kinetics of Plk1 activation before mitotic entry. One can postulate that the kinetics of Plk1 activation is perturbed after low doses of aphidicolin, which could trigger unscheduled separase/pericentrin phosphorylation in G2 (to be evaluated, see point 5) while not sufficient for mitotic entry. The authors should provide the kinetics of Plk1 activation in G2 in control versus low dose aphidicolin-treated cells (200nM and 400nM). To that purpose, they might take advantage of cells synchronized in G1 with CDK4/6i.

5) To quantitatively evaluate the lack of pericentrin organization around centrioles, the authors evaluate the % of cells with complete PCNT ring. According to the images provided in Figure 8A, the rings appear better formed after Aphidicolin treatment (upper right panels) than after siRNA of separase (bottom left panels), which is not supporting the quantitative analysis provided in Figure 8B, where centrosomes are not expected to be different from the control situation after separase siRNA alone. Is it again a misleading choice of images?

Can the authors provide an evaluation of the efficiency of the separase siRNA treatment by Western-blot? Can they perform the corresponding rescue experiment to further provide evidence of the specificity of the treatment?

6) The final schema in Fig8D shows direct activation of pericentrin and separase by Plk1-dependant phosphorylation based on the canonical PCM disassembly pathway described at mitotic exit. The authors do not provide experimental evidence in the manuscript that the same pathway is at play in G2. For example, they show that CEP57 is preserved after Aphidicolin treatment, indicating that centriole disengagement is differently achieved after unscheduled PCM disassembly. The authors should combine a Plk1 or CDK1 inhibitor and aphidicolin treatment and evaluate pericentrin and separase phosphorylation status as well as ring assembly in each condition. This experiment will help clarify the relative contribution of Plk1- and CDK1- dependent phosphorylation of pericentrin and/or separase in unscheduled PCM disassembly claimed in Figure 8.

7) In the manuscript, Fig2A provides a schema of replication stress/DNA damage pathways. To my sense, such a schema is useful in a review manuscript to summarize data published by different laboratories and in different model systems. The schema provided in the present manuscript should summarize only experimental data presented by the authors. I have the same remark for the model presented in Figure 8D, where each arrow or phosphorylation status should be demonstrated by the authors.

8) Minor remark: The movies related to Figure 3 clearly illustrate the presence or absence of transient multipolarity. Nonetheless, it is impossible to evaluate with the corresponding still merge images provided in Figure 3A. The authors should present EB3-GFP and H2B-mCherry channels separately in black and white.

Reviewer #2 (Remarks to the Author):

This is a very interesting paper that follows on from an earlier publication from this group showing that treatment of RPE1 cells with aphidicolin leads to premature centriole disengagement in mitosis. The authors now use expansion microscopy as a tool to examine the effects of aphidicolin treatment on centriole disengagement under a variety of conditions. The power of this assay is that it allows them to show that this centriole disengagement following aphidicolin treatment occurs much earlier in G2 and then use a series of pharmacological treatments to systematically inhibit components of the pathways downstream of either the ATR or ATM damage repair kinases. This reveals that inhibiting ATR, Chk1, and Wee1 kinases all prevent centriole disengagement. Paradoxically, even though the Wee1 inhibitor of Cdk1 is required, there remains a requirement for some Cdk1 activity, which they show is associated with cyclin A2, upstream regulator of Plk1, and not cyclin B1. Accordingly, there is a requirement for Plk1 and yet cells do not enter mitosis. This leads to the interesting hypothesis that differential levels of Plk1 activity are need for centriole separation as opposed ot mitotic entry, which the authors nicely test using a Plk1 FRET sensor. The premature separation observed appears to utilise the normal machinery for this process as separate depletion suppresses disengagement leaving an intact ring of preicentrin.

This is a very nice study. It is methodical in its systematic approach to studying the effects of disabling successive members of well-known pathways and provides insight into how Plk1 can have myriad functions in mitotic entry through the requirement for differential levels of kinase activity for its differing roles.

The final speculation that replication stress might lead to the accumulation of supernumerary centrioles seems highly likely and echoes findings by Raff and Glover over 30 years ago that inhibition of DNA replication with aphidicolin in *Drosophila* embryos allows the centriole duplication cycle to continue (J Cell Biol 1988 107:2009-19).

I strongly recommend publication of the manuscript without need for revision.

Reviewer #3 (Remarks to the Author):

Faithful DNA replication is critical for the maintenance of genomic integrity. Replication stress is linked to structural chromosomal aberrations. DNA and centrosome cycles asynchronization contributes to genomic instability. Devashish et al. claims that mild replication stress (400 nM APH) induces premature

centriole disengagement already in G2 via the ATR-Chk1 axis pathway, resulting sub-critical Plk1 kinase activity that primes the pericentriolar matrix for Separase-dependent disassembly.

Despite the reasonable and interesting of the paper, some issues are unconvincing and cause for concern: (1) Aphidocolin (APH) is a DNA polymerase inhibitor, treatment of cells with 200–400 nM of APH is a widely used to induce so-called mild replication stress. However, increasing phosphorylation of Chk1 and RPA could be detected in this concentration indicating cell cycle checkpoint activation already (PMID: 31448675). That suggest 400 nM APH treatment activates the ATR-CHK1 signaling pathway, which should occur in the S phase, but the centriole disengagement in the S phase mentioned in the article has not changed, so who drives and does it in the G2 phase? (2) HCT116 cells or non-transformed RPE-1 cells with 100 nM aphidicolin to induce mild replication stress during S phase, reduced replication fork progression, increased origin firing triggers microtubule dynamics and whole chromosome missegregation (PMID: 36516748). Is there a link among replication fork progression reduction, origin firing triggers microtubule dynamics and premature centriole disengagement in G2 by mild replication stress? Cause this preceding process occurring in S phase.

In addition, the novelty of the article is weakened. (1) Mild replication stress causes chromosome missegregation via premature centriole disengagement was reported before (PMID: 31395887). (2) Replication stress leads to transient spindle multipolarity and causes premature centriole disengagement which depends on the G2 activity of the Cdk, Plk1 and ATR kinases was reported before (PMID: 31395887). Therefore, the function of ATR-CHK1-Wee1-Cdk1/Cyclin-A/PLK1-Separase axis to regulate premature centriole disengagement is predictable and imaginable, and there is no novel attractive mechanism. (3) Due to the insufficient resolution of classical fluorescence microscopy in the previous study (PMID: 31395887), in this study the authors using expansion microscopy to demonstrate that mild replication stress already causes centriole disengagement in G2. However, this is an innovation in the experimental technology of fluorescence imaging, and no new breakthrough has been made in the molecular mechanism. (3) The ATR-Chk1 pathway plays a role in the generation of centrosome aberrations was reported in 2009 (PMID: 19403737). In this manuscript, there no new direct substrates were identified and the mechanism presented was not convincing. In conclusion, due to the above concerns, the reviewer does not recommend publication in this journal.

Comments:

(1) Line 127, “Chkk1i-2” should be “CHK1i-2”.

(2) Line 206, ref. 38-40, The choice of papers to cite throughout the manuscript should be reviewed carefully. In many cases reviews are cited instead of the primary papers that support a statement. This may be fine for general statements like ATR safeguards against replication stress, but not when a specific mechanistic discovery is described.

First, we would like to thank all three reviewers for the constructive input. We have addressed their concerns in the following manner.

Point-by point rebuttal

Reviewer #1 :

In the present manuscript, Dwivedi et al. provide further mechanistic insights into how mild replication stress causes premature centriole disengagement. The present manuscript relies on a previous publication by the Meraldi group where a clear link between mild replication stress, premature centriole disengagement and chromosome segregation errors was reported (Wilhelm et al., Nat Commun, 2019). Cdk1 and Plk1 requirement upstream of centriole disengagement have already been described in the previous manuscript. The main novelty of the manuscript by Dwivedi is the notion of a sub-critical level of Plk1 activity during the G2 phase of the cell cycle, unravelled by an elegant use of a Plk1 FRET biosensor. Restrained Plk1 activity level in G2 appears not sufficient for premature mitotic entry but sufficient to prime unscheduled separase-dependant PCM disassembly according to the authors. If this mechanistic explanation appears appealing, the authors need to provide more experimental evidences and clarify important points to fully support their model before the manuscript is suitable for publication. The manuscript is well written, referenced and presented, even if some clarifications are required.

We thank the reviewer for the positive comments and have addressed the concerns in the following way:

Here are my major concerns on the manuscript:

1) In the previous publication, premature centriole disengagement was noticed in early mitosis after conventional microscopy. In this manuscript, the authors observe premature centriole disengagement in G2 phase of the cell cycle after expansion microscopy technique. In Figure 1, the examples provided do not enable to clearly distinguish between normal and abnormal centriole configurations. For example, to my sense, centriole configurations appear similar in Fig1B and Fig1D after aphidicolin treatment, contrary to what is quantified in Fig1C and Fig1E, where 100% of centrioles appear engaged in S phase but only 55% in G2 phase. What shall we believe? Is it a misleading choice of images?

The fact that centriole disengagement does not occur before G2 phase is important to support the model proposed by the authors and more experiments are required to show that premature centriole disengagement is restricted to G2. The authors could synchronize the cells in G1 with CDK4/6 inhibitor before release in S phase with low dose aphidicolin treatment and analysis of centriole configurations in G2 phase (at least 8h post-release?).

We thank the reviewer for these comments. First, we have now chosen images that better reflect our quantifications in all panels of Figure 1. Second, as suggested by the reviewer, we also performed a synchronization experiment, by arresting cells at the G1/S transition with a Cdk4/6 inhibitor, releasing them for 4 or 8 hours in the presence of DMSO or low doses of Aphidicolin, and analysing centriole configuration by expansion microscopy. As shown in the new Figure 1G and H, low doses of Aphidicolin only lead to disengaged centrioles 8 hours after a G1/S release, confirming our conclusion that an Aphidicolin treatment does not lead centriole disengagement in S-phase.

2) In the great majority of examples provided, only one of the two centrosomes display premature centriole disengagement (one arrow in Fig1D, 2B, 3D...). How can the authors explain this asymmetry? Can they quantify it? Is the mother or daughter centrosome more prone to centriole disengagement? If this is the case, pericentrin disassembly driven by separase activity and responsible for centriole disengagement according to the authors should also be asymmetric. Nonetheless, the authors observe pericentrin disassembly on both centrosomes according to the images provided in Fig7, 8 and S3. How can the authors explain this apparent discrepancy?

We thank the reviewer for this interesting question that we had not considered originally. To address this point, we stained the G2 cells with CentrobIn, a protein known to localize at this stage on the daughter centrioles in each centrosome, and on the parental centriole of the young centrosome (LeRoux-Bourdieu, JCS, 2022). Using expansion microscopy, we again found that 40% of the cells contained at least one disengaged centriole pair. A more detailed analysis indicated that in 10% of the cells the old centrosome had dis-engaged its centrioles, in 14% the young centrosome has dis-engaged its centrioles, and in 15% of the cases both centrosomes displayed dis-engaged centrioles (Figure 1I and J). From these results, we conclude that a) there is no preference in terms of centriole dis-engagement between the old and the young centrosome; and b) that the probability that both centrioles pairs are dis-engaged is statistically linked: if one centriole pair is disengaged, the other centriole pair is more likely to also be disengaged than what would be predicted if both centrosomes were independent of each other. As elaborated in our manuscript, this observation is consistent with a common upstream origin for both events, which we postulate to be a prolonged intermediated Plk1 activity in G2.

3) The authors used a FRET biosensor to unravel a sub-critical level of Plk1 activity in G2 after low doses of aphidicolin. The notion of sub-critical level relies on a direct comparison of Plk1 activity level with G1 phase (lower level) and prometaphase (higher level). The authors provide only one value of Plk1 activity for the whole G2 phase without considering the kinetics of Plk1 activation before mitotic entry. One can postulate that the kinetics of Plk1 activation is perturbed after low doses of aphidicolin, which could trigger unscheduled separase/pericentrin phosphorylation in G2 (to be evaluated, see point 5) while not sufficient for mitotic entry. The authors should provide the kinetics of Plk1 activation in G2 in control versus low dose aphidicolin-treated cells (200nM and 400nM). To that purpose, they might take advantage of cells synchronized in G1 with CDK4/6i.

We thank the reviewer for the suggestion. As suggested by the reviewer we synchronized the cells in G1 with CDK4/6 inhibitor and monitored at different times after the release Plk1 activity with the FRET biosensor. Our quantification shows a small difference 4-8h hours after the release, but 10 hours after the G1 release, DMSO-treated cells fully activate Plk1 while cells experiencing mild replication stress only show intermediate Plk1 activity levels. These new quantitative data shown in Figure 5E and S3A-D, fully support our main conclusion that mild replication stress leads to an intermediate Plk1 activity that is sufficient to drive centriole disengagement, but insufficient to promote rapid mitotic entry.

5) To quantitatively evaluate the lack of pericentrin organization around centrioles, the authors evaluate the % of cells with complete PCNT ring. According to the images provided in Figure 8A, the rings appear better formed after Aphidicolin treatment (upper right panels) than after siRNA of separase (bottom left panels), which is not supporting the quantitative analysis provided in Figure 8B, where centrosomes are not expected to be different from the control situation after separase siRNA alone. Is it again a misleading choice of images?

Can the authors provide an evaluation of the efficiency of the separase siRNA treatment by Western-blot? Can they perform the corresponding rescue experiment to further provide evidence of the specificity of the treatment?

First, we apologize, both images had been swapped while assembling the figure. To be on the safe side, we replaced them with new examples. More importantly, we now also provide in Supplementary Figure S4D a Western-blot documenting an efficient separase depletion. Moreover, as suggested by the reviewer we carried out the separase rescued experiment. Given that such a rescue was not possible by transient transfection, we generated a stable RPE1 cell line expressing a Tet-On RNAi-resistant myc-Separase. We show that this construct rescues the lack of centriole disengagement in siSeparase-treated cells after mild replication stress; it does, however, not induce centriole disengagement on its own without replication stress (Figure 8F-H). These results thus validate our conclusion that centriole disengagement in G2 after mild replication stress is Separase-dependent.

6) The final schema in Fig8D shows direct activation of pericentrin and separase by Plk1-dependant phosphorylation based on the canonical PCM disassembly pathway described at mitotic exit. The authors do not provide experimental evidence in the manuscript that the same pathway is at play in G2. For example, they show that CEP57 is preserved after Aphidicolin treatment, indicating that centriole disengagement is differently achieved after unscheduled PCM disassembly. The authors should combine a Plk1 or CDK1 inhibitor and aphidicolin treatment and evaluate pericentrin and separase phosphorylation status as well as ring assembly in each condition. This experiment will help clarify the relative contribution of Plk1- and CDK1- dependent phosphorylation of pericentrin and/or separase in unscheduled PCM disassembly claimed in Figure 8.

We agree with the reviewer that our final scheme assumed some phosphorylation status that we had not directly tested. As suggested by the reviewer we tested whether Plk1 or Cdk1 inhibition in Aphidicolin-treated cells can rescue the pericentrin ring assembly and show that this is indeed the case, which is consistent with our original scheme. We could, however, not directly probe the phosphorylation status of separase. To our knowledge there is no phospho-antibody for the Plk1 site on separase (we consulted different experts in the field, such as Elmar Schiebel and Andreas Boland); all the published experiments were carried out with Separase mutants. Given that it already took us several months to create a stable cell line expressing wild-type Tet-On Separase for the rescue experiment, we feel that carrying out similar experiments with a Separase mutant is beyond the scope of this present study. Similarly, when we contacted Kunsoo Rhee to ask for the pericentrin phospho antibody, he informed us that his laboratory had run out of the purified antibody. We therefore removed the phosphorylation sites from our final scheme, showing only the elements that are supported by experimental evidence.

7) In the manuscript, Fig2A provides a schema of replication stress/DNA damage pathways. To my sense, such a schema is useful in a review manuscript to summarize data published by different laboratories and in different model systems. The schema provided in the present manuscript should summarize only experimental data presented by the authors. I have the same remark for the model presented in Figure 8D, where each arrow or phosphorylation status should be demonstrated by the authors.

To address this concern, we removed the schema in Figure 2 and present a modified scheme in Figure 9 (see also point 6)

8) Minor remark: The movies related to Figure 3 clearly illustrate the presence or absence of transient multipolarity. Nonetheless, it is impossible to evaluate with the corresponding still merge images provided in Figure 3A. The authors should present EB3-GFP and H2B-mCherry channels separately in black and white.

As suggested by the reviewer, we now present the channels separately in Figure 3A.

Reviewer #2 (Remarks to the Author):

This is a very interesting paper that follows on from an earlier publication from this group showing that treatment of RPE1 cells with aphidicolin leads to premature centriole disengagement in mitosis. The authors now use expansion microscopy as a tool to examine the effects of aphidicolin treatment on centriole disengagement under a variety of conditions. The power of this assay is that it allows them to show that this centriole disengagement following aphidicolin treatment occurs much earlier in G2 and then use a series of pharmacological treatments to systematically inhibit components of the pathways downstream of either the ATR or ATM damage repair kinases. This reveals that inhibiting ATR, Chk1, and Wee1 kinases all prevent centriole disengagement. Paradoxically, even though the Wee1 inhibitor of Cdk1 is required, there remains a requirement for some Cdk1 activity, which they show is associated with cyclin A2, upstream regulator of Plk1, and not cyclin B1. Accordingly, there is a requirement for Plk1 and yet cells do not enter mitosis. This leads to the interesting hypothesis that differential levels of Plk1 activity are needed for centriole separation as opposed to mitotic entry, which the authors nicely test using a Plk1 FRET sensor. The premature separation observed appears to utilise the normal machinery for this process as separate depletion suppresses disengagement leaving an intact ring of precentrin.

This is a very nice study. It is methodical in its systematic approach to studying the effects of disabling successive members of well-known pathways and provides insight into how Plk1 can have myriad functions in mitotic entry through the requirement for differential levels of kinase activity for its differing roles.

The final speculation that replication stress might lead to the accumulation of supernumerary centrioles seems highly likely and echoes findings by Raff and Glover over 30 years ago that inhibition of DNA replication with aphidicolin in *Drosophila* embryos allows the centriole duplication cycle to continue (J Cell Biol 1988 107:2009-19).

I strongly recommend publication of the manuscript without need for revision.

We thank the reviewer for the positive comments and for pointing out the findings in flies by the Glover laboratory, which we now cite in our discussion.

Reviewer #3 (Remarks to the Author):

Faithful DNA replication is critical for the maintenance of genomic integrity. Replication stress is linked to structural chromosomal aberrations. DNA and centrosome cycles asynchronization contributes to genomic instability. Devashish et al. claims that mild replication stress (400 nM APH) induces premature centriole disengagement already in G2 via the ATR-Chk1 axis

pathway, resulting sub-critical Plk1 kinase activity that primes the pericentriolar matrix for Separase-dependent disassembly.

Despite the reasonable and interesting of the paper, some issues are unconvincing and cause for concern: (1) Aphidocolin (APH) is a DNA polymerase inhibitor, treatment of cells with 200–400 nM of APH is a widely used to induce so-called mild replication stress. However, increasing phosphorylation of Chk1 and RPA could be detected in this concentration indicating cell cycle checkpoint activation already (PMID: 31448675). That suggest 400 nM APH treatment activates the ATR-CHK1 signaling pathway, which should occur in the S phase, but the centriole disengagement in the S phase mentioned in the article has not changed, so who drives and does it in the G2 phase? (2) HCT116 cells or non-transformed RPE-1 cells with 100 nM aphidicolin to induce mild replication stress during S phase, reduced replication fork progression, increased origin firing triggers microtubule dynamics and whole chromosome missegregation (PMID: 36516748). Is there a link among replication fork progression reduction, origin firing triggers microtubule dynamics and premature centriole disengagement in G2 by mild replication stress? Cause this preceding process occurring in S phase.

In addition, the novelty of the article is weakened. (1) Mild replication stress causes chromosome mis-segregation via premature centriole disengagement was reported before (PMID: 31395887). (2) Replication stress leads to transient spindle multipolarity and causes premature centriole disengagement which depends on the G2 activity of the Cdk, Plk1 and ATR kinases was reported before (PMID: 31395887). Therefore, the function of ATR-CHK1-Wee1-Cdk1/Cyclin-A/PLK1-Separase axis to regulate premature centriole disengagement is predictable and imaginable, and there is no novel attractive mechanism. (3) Due to the insufficient resolution of classical fluorescence microscopy in the previous study (PMID: 31395887), in this study the authors using expansion microscopy to demonstrate that mild replication stress already causes centriole disengagement in G2. However, this is an innovation in the experimental technology of fluorescence imaging, and no new breakthrough has been made in the molecular mechanism. (3) The ATR-Chk1 pathway plays a role in the generation of centrosome aberrations was reported in 2009 (PMID: 19403737). In this manuscript, there no new direct substrates were identified and the mechanism presented was not convincing. In conclusion, due to the above concerns, the reviewer does not recommend publication in this journal.

We thank the reviewer for these constructive concerns, which we have addressed in the following manner:

- a) **Is the premature centriole disengagement due to events occurring in S-phase?** Our experiments in Figure 1 indicate that mild replication stress in S-phase is not sufficient to induce centriole dis-engagement. By performing cell synchronization experiments (point 1 of reviewer 1; Figure 1G and H) and monitoring Plk1 activity during S and G2 phase (point 3 of reviewer 1; Figure 5E and S3A-D), we now show that mild replication stress only induces a major difference in Plk1 activity in the context of a G2 cell, allowing it to drive centriole disengagement but not mitotic entry. These new experiments thus validate our conclusion that premature centriole disengagement is specific for G2 and does not occur in S-phase, and they provide a dynamic and novel molecular mechanism (differential Plk1 activity) for this phenomenon.
- b) **Is the phenotype linked to the change in microtubule dynamics described by the Bastians laboratory (PMID36516748)?** Yes, it is true that we have reported that mild replication stress can induce changes in mitotic microtubule dynamics (see PMID

31395887), a fact that was later also reported by the Bastians laboratory (PMID36516748 and PMID 31448675). These reports, however, only found a change in the dynamics of mitotic microtubules, and did not report changes in G2, when centriole disengagement has already occurred, as we now show. Our own study (PMID 31395887) shows that the change in microtubule dynamics contributes to the formation of aberrant spindle, reinforcing the effects of centriole disengagement; nevertheless, we also show that in a number of colorectal cancer cell lines (HCC70 and HCC1187), mild replication stress is not associated to change in microtubule dynamics, yet still drives premature centriole disengagement. We therefore concluded that changes in microtubule dynamics are the primary cause for centriole disengagement.

- c) **Novelty of the presented findings:** here, we respectfully disagree with the reviewer. Yes, we had previously shown that mild replication stress caused premature centriole disengagement in mitosis and required Cdk1, Plk1, and ATR activity. However, when exactly centriole disengagement (S-phase, G2, immediately prior to NEBD) was unknown and an open question (as shown by point 1 of reviewer 3) and the comments of this reviewer, who suggests that these events could already happen in S-phase. Second, yes based on our initial findings one could have predicted that the ATR-CHK1-Wee1-Cdk1/Cyclin-A/PLK1-Separase is implicated, but and experimentally proving a point is still essential. In fact, one could have also predicted that this pathway would involve Cep57, yet it does not. Therefore, identifying the relevant downstream targets of Plk1 was answering a key question. Third, and most important, our previous data raised a paradox: how could centriole disengagement both depend on Plk1 activity and on ATR activity, given that ATR suppresses Plk1 activity? We believe that our results indicating that mild replication stress does not allow a full activation of Plk1 in late G2, but instead induces an intermediate Plk1 activity that is sufficient for centriole disengagement but not for rapid mitotic entry provides an important and non-intuitive response to this paradox. This is the central concept of this manuscript, identifying a novel molecular mechanism that can disrupt the synchrony of the DNA and centrosome cycle and thus contribute to genetic instability in cancer cells.

Comments:

- (1) Line 127, “Chkk1i-2” should be “CHK1i-2”.

We thank the reviewer for this point, which we have corrected.

- (2) Line 206, ref. 38-40, The choice of papers to cite throughout the manuscript should be reviewed carefully. In many cases reviews are cited instead of the primary papers that support a statement. This may be fine for general statements like ATR safeguards against replication stress, but not when a specific mechanistic discovery is described.

We thank the reviewer for this suggestion. We have now included more primary publications.

REVIEWERS' COMMENTS

Reviewer #1 (Remarks to the Author):

I consider that the authors have fully addressed my concerns, by providing new experimental evidence to support their model or by discussing appropriately some critical points. I now support the manuscript for publication without further revision.

Reviewer #3 (Remarks to the Author):

The author has addressed all my concerns and the article is acceptable for publication after this revision.